

# Reconstruction and Projection of Sea Level Around the Korean Peninsula Using Cyclostationary Empirical Orthogonal Functions

Se-Hyeon Cheon[12], Benjamin D. Hamlington[1], Kyung-Duck Suh[23]

[1]Department of Ocean, Earth and Atmospheric Sciences, Old Dominion University, Norfolk, VA 23529, USA
[2]Department of Civil Engineering, Seoul National University, Seoul, 08826, South Korea
[3]Institute of Engineering Research and Entrepreneurship, Seoul National University, Seoul 08826, Republic of Korea

*Correspondence to*: Benjamin D. Hamlington (bhamling@odu.edu)

**Abstract.** Since the advent of the modern satellite altimeter era, the understanding of the sea level has increased dramatically. The satellite altimeter record, however, dates back only to the 1990s. The tide gauge record, on the other hand,
extends through the 20th century, but with poor spatial coverage when compared to the satellites. Many studies have been conducted to extend the spatial resolution of the satellite data into the past by finding novel ways to combine the satellite data and tide gauge data in what are known as sea level reconstructions. However, most of the reconstructions of sea level were conducted on a global scale, leading to reduced accuracy on regional levels, particularly where there are relatively few tide gauges. The sea around the Korean Peninsula is one such area with few tide gauges prior to 1960. In this study, new
methods are proposed to reconstruct the past sea level and project the future sea level around the Korean Peninsula. Using spatial patterns obtained from a cyclo-stationary empirical orthogonal function decomposition of satellite data, we reconstruct sea level over the time period from 1900 to 2014. Sea surface temperature data and altimeter data are used simultaneously in the reconstruction process, leading to an elimination of reliance on tide gauge data. Although the tide gauge data was not used in the reconstruction process, the reconstructed results showed better agreement with the tide gauge
observations in the region than previous studies that incorporated the tide gauge data. This study demonstrates a reconstruction technique that can be used on regional levels, with particular emphasis on areas with poor tide gauge coverage.

## 1 Introduction

Although sea level rise is a global phenomenon, the impacts are local, and are happening now. Changes in sea level are
impacting communities across the globe on an almost daily basis through increased erosion, greater saltwater intrusion, more frequent "nuisance" flooding, and higher storm surge (e.g. Nicholls, 2011; Cheon and Suh 2016, Suh et al. 2013). Planning for, adapting to, and mitigating current and future sea level has necessarily begun in many threatened areas. Expensive decisions - both in economic and societal terms - are already being made. Examples can be found throughout the world, with coastal communities making difficult decisions on how to address concerns associated with future sea level rise (e.g.
Nicholls, 2011). The present and near-term threat of sea level rise across the globe and the subsequent decisions to address





the problem highlight the immediate need for actionable regional sea level projections across a range of actionable timescales.

Before the satellite altimeter era, the only available sea level observations came from tide gauge (after this TG) records. The TG data provide records of local sea level variations, covering a time period of nearly two hundred years in some locations around the globe. Using TG data, scientists can study past sea level changes at specific locations across the globe. However, TGs do not provide good global coverage as they are necessarily only located at coastal sites and have a bias towards the Northern Hemisphere. Satellite altimeters, on the other hand, have been collecting data since 1992. The satellite altimetry data have near-global coverage of sea level but a relatively short observation period compared to TG observations, which is a severe handicap to analyzing long-term changes in sea level. This disadvantage is particularly true given the presence of sea level variability with decadal timescales.

Chambers et al. (2002) attempted to reconstruct sea level anomalies (SLA) by combining TG data and satellite altimeter data. In their research, they studied low-frequency variability in global mean sea level (or global mean sea level anomaly; hereafter GMSL or GMSLA) from 1950 to 2000. They interpolated sparse TG data into a global gridded SLA pattern applying EOFs (Empirical Orthogonal Functions) of SLA using data from the TOPEX/Poseidon satellite altimeter to capture the interannual-scale signals, e.g., El Nino-Southern Oscillation (hereafter ENSO) and the Pacific Decadal Oscillation (from now on PDO). Building on previous studies (Chambers et al., 2002; Kaplan et al., 1998; Kaplan et al., 2000), Church et al. (2004) created a reconstruction from 1950 to 2001 using EOFs of SLA data measured from satellite altimeter and a reduced space optimal interpolation scheme. This research was subsequently updated to increase temporal coverage from 1870 to the present (Church and White et al., 2006, 2011) and the reconstructions have been made available to the public. In these studies, GMSL was found to rise approximately 210 mm from 1880 to 2009, with a linear trend from 1900-2009 of $1.7 \pm 0.2$ mm per year. The resulting SLA is one of the most comprehensive and widely cited reconstructions. While these studies focused largely on the reconstruction of GMSL, Hamlington et al. (2011) applied cyclostationary empirical orthogonal functions as basis functions for the reconstruction of SLA in an attempt to improve the representation of variability about the long-term trends. This approach has been shown to provide an advantage for describing local variations such as ENSO and PDO. After that, Hamlington et al. (2012a) proposed an improved scheme of their reconstruction using sea surface temperature (hereafter SST). Given the limited TG data in the past, reconstructions of sea level relying only on TGs are poor, particularly before 1950. Leveraging other ocean observations (e.g. SST) to reconstruct sea level leads to an improved sea level reconstruction further into the past.

While sea level is a global phenomenon, the extent of sea level change can vary dramatically across the globe. During the 24-year satellite altimeter record, regional trends have been measured to be four times greater than the global average in some areas. Properly planning for future sea level change requires an assessment of sea level on local or regional levels, as future sea level for one location could be quite different than future sea level in another location. Rather than using a global reconstruction, several studies have instead focused on regional reconstructions of sea level, targeting a specific area of



focus. As an example, using an optimal interpolation method, Calafat and Gomis (2009) reconstructed the distribution of SLA in the Mediterranean Sea over 1945-2000. They used EOFs of satellite altimeter data spanning 1993-2005 as basis functions and interpolated the TG data using these spatial patterns. A spatial distribution of sea level rise trends for the Mediterranean for the period of 1945-2000 indicated a positive trend in most areas. Hamlington et al. (2012b) performed a

regional sea level reconstruction based on the scheme applying CSEOFs (Hamlington et al., 2011) with a domain covering only the Pacific Ocean. They found that a choice of basis functions had a significant effect on the spatial pattern of the sea level rise and the ability to capture internal variability signals. Global basis functions – either CSEOF or EOF – are typically dominated by large-scale variability in the Pacific Ocean associated with ENSO or the PDO. As a result, global reconstructions are poorer in some ocean basins (Indian Ocean, Atlantic Ocean) than others (Pacific Ocean). This issue is

likely exacerbated even further when looking at even smaller regions.

In this paper, we focus on one such region – the Korean Peninsula where over seventy-five million people live. In South Korea, over twenty-seven percent of people live in coastal city areas, and nearly thirty-six percent of GRDP (Gross Regional Domestic Product) is produced by coastal city regions. As a result, policymakers have a keen interest in a sea level rise around the Korean Peninsula (hereafter KP) to establish proper remedies to future sea level rise. Using global reconstructions

around the KP, or more generally any small region, is a problem. First, global scale reconstructions use a limited number of basis functions to prevent interpolation from over-fitting and creating spurious sea level fluctuations. There is a difference between the major modes for global scale and the major modes for local scale; e.g. there is a high possibility that the globally selected basis functions, which represent 90 % of the total variance in the global level, for example, will not represent 90 % of the total variance in local scale. Second, the temporal coverage of the TG data around the KP started around 1930 when

less than 10 TGs were available; it is too little to secure accuracy on these local scales. As mentioned above, TG coverage is poor extending back into the 20[th] century, and looking at the regional level will lead to relatively few gauges to analyse in most areas. Hence, the goal of this study is proposing a new scheme that builds off of Hamlington et al. (2012a) that applies CSEOFs to reconstruct local SLA where the TG data is not enough to ensure a quality reconstruction through the 20[th] century. We focus on the KP both due to its exposure to risk from impending sea level rise and also as a test case to

demonstrate how this technique could be applied at other locations across the globe. In brief, the primary goals of this study can be summarized as follows: 1) Broaden our understanding of the SLA around the KP both in the past and present and 2) Suggest a new reconstruction scheme for local areas where have insufficient tide gauge coverage in spatial and temporal domain.





## 2 Data and Methods

### 2.1 Data

#### 2.1.1 Sea level anomaly

The basis functions for this reconstruction are the CSEOFs monthly mean gridded SLA covering the time period from 1993

to present. This monthly data has a $0.25° \times 0.25°$ grid resolution and it is available via the AVISO (the Archiving, Validation, and Interpretation of Satellite Oceanographic); this data opens in public (ftp://ftp.aviso.altimetry.fr/global/delayed-time/grids/climatology/monthly_mean/), and hereafter this data set is written as AVISO-SLA. The data is based on satellite altimeter measurements over 1993-2015; Topex/Poseidon, ERS-1&2, Geosat Follow-On, Envisat, Jason-1&2, and OSTM satellites collected the SLA. The delayed time Ssalto/DUACS multi-mission

altimeter data processing system has created this product. Before conducting the CSEOF decomposition, mean values for each grid point were removed to center the data. The annual signal has not been removed as it is accounted for by the CSEOF analysis (see more details in section 2.2.1 below). The data was trimmed to contain only the ocean around the KP (31°-46°N and 117°-142°E; hereafter AVISO-KP) and it was multiplied by the square root of the cosine of latitude to consider the actual area of each grid.

**2.1.2 Sea surface temperature**

In this study, two SST reconstruction data sets were used: ERSST (Extended Reconstructed Sea Surface Temperature) (Huang et al., 2015a; Huang et al., 2015b; Liu et al., 2015) and COBESST2 (Centennial in situ Observation-Based Estimates; Ishii et al. (2005)). The ERSST dataset is a global monthly SST dataset based on the observation of ICOADS (International Comprehensive Ocean–Atmosphere Dataset). This monthly analysis has a $2° \times 2°$ grid resolution and its time

coverage is from 1854 to the present, and the include data are anomalies based on a monthly climatology computed from 1971–2000. The COBESST2 dataset is a monthly interpolated $1°x1°$ SST product over 1850 to the present. It integrates several SST observations: ICOADS 2.5, satellite SST, and satellite sea ice. The bucket correction process was applied to the data up to 1941. In addition to OI (Optimal Interpolation) scheme, this data set used an EOF reconstruction.

    Each data was trimmed as three different regions: a global domain (no trim), the Northwest Pacific (NWP) domain (25°-

55°N and 110°-160°E), and around the KP area; to indicate the domains of dataset we put '-NWP' and '-KP' behind the name of dataset. Before conducting the CSEOF decomposition, these data sets were treated as follows. 1) The mean values for each grid point were removed to prevent those values to have a significant influence on CSEOFs. 2) The data were weighted by the square root of the cosine of latitude to consider the actual area of each grid. 3) Any grid points that were not continuous in time were removed. Like the satellite altimeter dataset, the annual signal of SST data was not removed.





### 2.1.3 Tide gauge data

Monthly mean sea level records of 47 TGs for the KP were obtained from the Permanent Service for Mean Sea Level (hereafter PSMSL, see Fig. 1) over 1930-2013. The Revised Local Reference (RLR) data were selected; the RLR data are measured sea levels at each site about a constant local datum over the complete record. The earliest data of TG-KP is traced

back to 1930 at Wajima Station (see Fig. 1). Before 1965, the number of available TG datasets is fewer than 10, with only one TG (Wajima Station) providing data before 1950.

An ongoing GIA (glacial isostatic adjustment) correction was applied to the TG data using ICE-5G VM2 model (Peltier, 2004). Since an IB (Inverted Barometer) correction was applied to the satellite altimetry data, the TGs-KP data are IB-corrected based on the pressure fields from 20th Century Reanalysis V2c data (Compo et al., 2006; Compo et al., 2011;

Hirahara et al., 2014; Whitaker et al., 2004). The TG-KP in this study is modified with further editing criteria. The techniques for editing are similar to those of Hamlington et al. (2011), with TG-KP that have shorter record length than 5 years and unphysical trends (greater than 7 mm/yr) likely owing to uncorrectable vertical land motions being removed prior to analysis. After calculating a month-to-month change, jumps greater than 250 mm were also removed.

### 2.1.4 Reconstructed sea level of previous study

Church and White et al. (2006; 2011) created a reconstruction from 1870 to 2009 using EOFs of SLA from satellite altimeter over 1993-2009. They applied the Reduced Space Optimal Interpolation technique. According to their research, the GMSL rose about 210 mm over 1880-2009, and the linear trend through 1900-2009 was $1.7 \pm 0.2$ mm per year. The resulting SLA is one of the most comprehensive and widely cited reconstruction results. This data set was employed for long-term background trend for this study (see more detail below). The GMSL portion of this reconstruction has been extended and

made publicly available (http://www.cmar.csiro.au/sealevel/GMSL_SG_2011_up.html). To create the reconstructed sea level anomaly (hereafter ReSLA), Hamlington et al. (2011) combined the CSEOFs of the satellite altimetry and historical TG record. This approach provides an advantage for describing local variations such as ENSO and PDO. This weekly analysis has a $0.5° \times 0.5°$ grid resolution and its time coverage is over 1950-2009. This data set was used for the comparison with the reconstruction of this study. This reconstruction dataset (Hamlington et al., 2014) can be downloaded from a NASA

JPL/PO.DAAC (ftp://podaac.jpl.nasa.gov/allData/recon_sea_level/preview/L4/tg_recon_sea_level/).

### 2.2 Methods

Most of the studies on the reconstruction of sea level have been done on a global scale (Church and White, 2006, 2011; Church et al., 2004; Hamlington et al., 2012a; Hamlington et al., 2011; Hay et al., 2015). In some parts of the world with sparse observations, however, the quality of the reconstruction is poor. Hence to get more accurate results, a local scale

study is necessary to produce the level of quality that is necessary for planning and policy-making purposes. To date, this has





been an understudied area, however, with relatively few studies on the subject (Calafat and Gomis, 2009; Calafat and Jordà, 2011; Hamlington et al., 2012b).

The main difficulties are the lack of historical observations and poor spatial distributions of the TG data. The regional reconstruction of sea level around the KP suffers from these problems. The longest TG record extends back only to 1930, and most of the TG data is available only after the mid-1960s with relatively few available in the northern area of the KP. If previous reconstruction schemes are applied that rely only on sea level, then it is likely only possible to obtain reliable results after 1970. A modified reconstruction method is proposed for an area such as the KP having poor TG coverage. The approach is based on the CSEOF decomposition and multivariate regression while taking into account a time lag. This approach is a progression from the technique described in Hamlington et al. (2012a). In that study, given the relatively large region of reconstruction (Pacific Ocean basin), tide gauge observations were available for the entirety of the reconstructed record. In this case, suitable tide gauge coverage around the KP is only really available after the mid-1960s, necessitating an approach that is independent of the tide gauge observations. In this section, the procedure of the proposed scheme and fundamental theories are shown.

### 2.2.1 Cyclostationary empirical orthogonal functions

To understand the complex response of a physical system, the decomposition of data into a set of basis functions is frequently applied. The decomposed basis functions have the potential to give a better understanding of complex variability of the fundamental phenomenon. The simplest and most common computational basis functions are EOFs, which have often served as the basis for climate reconstructions. When a reconstruction selects the EOFs as basis functions, one basis function is defined as a single spatial map accompanied by a time series representing the amplitude modulation of this spatial pattern over time. The EOF decomposition of the spatio-temporal system, $T(r, t)$, is defined by the Eq. (1):

$$T(r, t) = \sum_i LV_i(r)PCT_i(t), \tag{1}$$

where $LV(r)$ is a physical process (or loading vector) modulated by a time series $PCT(t)$ (principal component time series or PC time series). Combining each LV and PCT pair, a signal of single EOF mode can be produced.

The assumption underlying EOF-based reconstruction is the stationarity of the spatial pattern represented by the EOF over the entire period. However, the fact that many geophysical phenomena are cyclostationary is well known (Kim et al., 2015). That is, these processes are periodic over a certain inherent timescale, with the amplitude of this periodic process varying over time. Even though EOFs represent cyclostationary signals through a superposition of multiple modes, as stated in Dommenget and Latif (2002), representing the cyclostationary signals with stationary EOFs can lead to an erroneous and ambiguous interpretation of the data. It also requires many EOFs to explain a relatively small amount of variability in a dataset.





To remedy some of these issues, Hamlington et al. (2011) introduced CSEOFs as the basis for SLA's reconstruction instead of EOFs. The CSEOF analysis has been proposed to capture the cyclo-stationary patterns and longer scale fluctuations in geophysical data (Kim and Chung, 2001; Kim et al., 2015; Kim and North, 1997; Kim et al., 1996; Kim and Wu, 1999). The CSEOF analysis can capture the time varying signals as a single mode by giving a time dependency to the loading vectors.

The system is defined as Eq. (2) and (3).

$$T(r,t) = \sum_i CSLV_i(r,t) PCT_i(t) \qquad (2)$$

$$CSLV(r,t) = CSLV(r,t+d) \qquad (3)$$

where $CSLV(r,t)$ is a cyclo-stationary LV and it is time dependent and periodic with a particular period $d$ (called a "nested period" and more details in the following sections). The studies of Kim et al. (1996), Kim et al. (1997) and Kim et al. (2015) provide more detailed walk-through for the CSEOF computation and properties.

CSEOFs provide significant advantages over EOFs since CSEOFs can explain cyclostationary signals in one mode; this means the opportunity of separating physical signals into a single, easy-to-interpret mode (Kim et al., 2015; Kim and North, 1997; Kim et al., 1996; Kim and Wu, 1999). Hamlington et al. (2011, 2012a, and 2012b) demonstrated that CSEOFs provided significant benefits dealing with repeating signals such as ENSO (El Niño–Southern Oscillation) and MAC (Modulated Annual Cycle) signals.

**2.2.2 Multivariate regression using CSEOFs**

When considering the complete Earth climate system, one variable is often directly connected to another variable. In some cases, they are impacted by a common physical process, or in other cases, one variable may directly influence another. To take advantage of these relationships and establish links, we can perform a multivariate linear regression as following Eq. (4).

$$y = \beta_0 + \beta_1 x_1 + \beta_2 x_2 + \cdots + \beta_k x_k + \varepsilon \qquad (4)$$

where $\beta_0, \beta_1, \beta_2, \cdots, \beta_k$ are regression coefficients and the $\varepsilon$ is random error. In this study, the response variables are each PCT of AVISO-KP's CSEOF and the predictor variables are all PCT of each SST dataset's CSEOF. Eq. (4) can be re-written as follows:

$$PCT_{SLA}^m = \beta_0^m + \beta_1^m PCT_{SST}^1 + \beta_2^m PCT_{SST}^2 + \cdots + \beta_k^m PCT_{SST}^k + \varepsilon^m \qquad (5)$$

where $PCT_{SLA}^m$ is the $m$-th PCT of SLA-KP's CSEOF and $\beta_k^m$ $\beta_0^m, \beta_1^m, \beta_2^m, \cdots, \beta_k^m$ are regression coefficients for the $m$-th target ($m = 1, 2, \cdots, M$; $M$ is total number of target's modes), and $PCT_{SST}^k$ is the $k$-th PCT of SST's CSEOF.

The matrix form of the Eq. (5) is:



$$\begin{bmatrix} T_1^m \\ T_2^m \\ \vdots \\ T_n^m \end{bmatrix} = \begin{bmatrix} 1 & P_1^1 & P_1^2 & \cdots & P_1^k \\ 1 & P_2^1 & P_2^2 & \cdots & P_2^k \\ \vdots & \vdots & \vdots & \ddots & \vdots \\ 1 & P_n^1 & P_n^2 & \cdots & P_n^k \end{bmatrix} \begin{bmatrix} \beta_0^m \\ \beta_1^m \\ \vdots \\ \beta_k^m \end{bmatrix} + \varepsilon^m \tag{6}$$

where $T_n^m$ is the $n$-th component of $PCT_{SLA}^m$, $P_n^k$ is the $n$-th component of $PCT_{SST}^k$. Additionally, many geophysical signals have lagged relations with other geophysical signals (Bojariu and Gimeno, 2003; Dettinger et al., 1998; Hamlet et al., 2005; Hendon et al., 2007; Kawamura et al., 2004; McPhaden et al., 2006; Redmond and Koch, 1991). Hence we think that the

PCTs which are mathematically independent of each other also can have a lagged relationship. If we consider the lagged relationships between the target and predictor variables and use the predictors having a higher correlation, we can reduce the number of predictors in the regression; generally, the more predictors applied for the regression, the more noise is likely to appear in the simulation. Before performing the multivariate linear regression system as in (5), we calculated the cross-correlation between the target PCT of SLA-KP and predictor PCTs of SST. The predictors were selected based on their

cross-correlation values. The threshold cross-correlation value did not have a sensitive effect on the regression if the value can select more than ten predictors; in the study, we used 0.3 as the threshold. By assuming the lag of the $i$-th mode having maximum cross-correlation at lag $\rho_i$, the $m$-th mode's PCT of AVISO-KP can be given as follow based on the Eq. (5).

$$PCT_{SLA}^m(t) = \beta_0^m + \sum_{i=1}^k \beta_i^m PCT_{SST}^i(t - \rho_i) + \varepsilon^m \tag{7}$$

### 2.2.3 Reconstruction of the past SLA-KP

By extending the PCT of AVISO-KP's CSEOFs, we can reconstruct the past SLA-KP. A unique characteristic of this reconstruction in contrast with others is the non-use of the local TG data sets. As mentioned above, the main motivation for this is the poor coverage of TGs around the KP. After removing GMSLA from the AVISO-KP at each grid point, the CSEOF decomposition was conducted. This means that if we conduct the reconstruction using AVISO-KP that has no GMSLA (hereafter AVISO-KP0), then the reconstructed SLA-KP0 (hereafter ReSLA-KP0) similarly includes no GMSLA

signal.

Using the regression coefficients and lagged relationship between the PCTs of each SST dataset and AVISO-KP0, we can extend the PCTs of AVISO-KP0 through Eq. (7). By combining the LVs of AVISO-KP0 and extended PCTs, we can rebuild the past SLA-KP albeit with no GMSLA. Finally, after adding the GMSLA to the ReSLA-KP with no GMSLA, the SLA-KP can be reconstructed with a regional mean sea level change.

To estimate the confidence intervals of the reconstructions in this study, both AVISO-KP and the SST reanalysis data are assumed as correct values. Based on the assumption, the multiple linear regression provides confidence intervals for each regression coefficient. A MC (Monte Carlo) simulation was carried out using the confidence intervals of multiple linear regression coefficients and GMSL. The MC simulation created 1000 sample-sets for ReSLA-KP with no GMSLA (hereafter ReSLA-KP0). By analysing 1000 sample-sets, we estimate the confidence interval of ReSLA-KP0. However, to ReSLA-



KP0 we need to add the GMSLA which has their own uncertainties. We used the GMSLA of Church and White (2011) which played the role of the long-term background change of the SLA-KP and this data-set provided their confidence intervals. Consequently, the overall confidence intervals of the current reconstruction can be estimated by summing the two confidence intervals.

A procedure of the current reconstruction can be summarized as following. Every SST dataset was trimmed to have the time span of 1891-2014. The AVISO-SLA was trimmed to contain only the data around the KP (31°-46°N and 117°-142°E). The southeast sea of the Japanese islands was removed.  Every SST dataset was cut into three regions: around the KP (same box with AVISO-KP; hereafter add '-KP'), the Northwest Pacific Ocean (N25°-55° and E110°-160°; hereafter add '-NWP'), and global (no trimming). All grid points that were not continuous in time were removed for every dataset. In total, we tested

six different SST data combinations. GMSLA and mean values were removed from AVISO-KP at each grid point. Each dataset was weighted by the square root of the cosine of latitude to consider the actual area of each grid. The CSEOF decomposition was applied to all data sets (AVISO-KP0 and SST datasets) with twelve months nested period. The lagged relation between PCTs of AVISO-KP0 and PCTs of each SST dataset were estimated with two years maximum lagging boundary. Using the PCTs of each dataset's CSEOF, the multiple linear regression systems were built based on Eq. (7) over

1993-2014. In this regression, the target variables were each PCT of AVISO-KP0 and the predictors are PCTs of each SST dataset. The regression coefficients and their confidence intervals were estimated to extend the target variables. Applying MC simulation that used the confidence intervals of regression coefficients, we randomly generated a thousand sample-sets of each extended PCT of AVISO-KP0. By combining the extended PCTs to the LVs of AVISO-KP0, we produced a thousand ReSLA-KP0s. By adding the GMSLA (Church and White, 2011) to the ReSLA_KP0s, a thousand of ReSLA-KPs

were generated. Finally, by statistical analysis of each time step of the random samples, we estimated the mean variation and their confidence intervals of each reconstruction.

For comparison, in addition to the TGs-KP, we used the reconstructed dataset of Hamlington et al. (2011); hereafter ReSLA-H. Their reconstruction was based on the TG records and satellite altimetry's CSEOF. The reconstruction results over 1970-2009 are quite reliable, because, after 1970, the number of available TG record around the world is enough to

guarantee the reconstruction results. The correlation coefficient ($\rho$) and NRMSE (Normalized Root Mean Square Error; we obtain this value through dividing RMSE by the standard deviation of the reference dataset; see Eq. 8) values for the entire domain and each TG location were calculated. By using these two values, we decided the best reconstruction case among the six reconstructions which are introduced in section 3.2.

$$NRMSE = 1 - \frac{\|x_{ref}(i) - x(i)\|}{\|x_{ref}(i) - \mu_{x_{ref}}\|} \tag{8}$$

where $\| \blacksquare \|$ indicates the 2-norm of a vector, $x_{ref}$ and $x$ are reference data and tested data respectively.



## 3 Results and Discussions

### 3.1 Sea Level Anomaly around the KP

Using AVISO-KP over 1993-2015, a linear trend map was estimated as shown in Fig. 2. The mean trend was found to be 3.1 ± 0.5 mm/yr. The linear trend of mean SLA-KP (hereafter MSLA-KP) agrees closely with the global SLA trend, 3.0 ± 0.0

mm/yr (see Fig. 3). Due to the similarity between the long-term trends of MSLA-KP and GMSLA, it is reasonable that the MSLA-KP can be described as the combination between background signals (GMSLA) and variabilities from the background signals (see Fig. 3). Most of the SLA-KP trends were close to the mean, but some parts of the East/Japan Sea, and of the Yellow Sea close to land, exhibited extreme patterns. Some areas showed trends over 7 mm/yr, while in other regions there were trends less than 1 mm/yr of the linear trend (see Fig. 2). To reason whether the extreme trends patterns

was related to the local mass distribution caused by various sources such as vortex and river discharge or was an independent phenomenon, we calculated the mean correlation $\rho$ (hereafter $\bar{\rho}$) of each AVISO-KP's grid point. For example, $\bar{\rho}$ at a single grid point $P$ was calculated by taking mean of $\rho$ values that had been estimated between $P$ and all other points. By repeating these calculations at all the points, we obtained Fig. 4. We deemed that the SLA of the regions having relatively high $\bar{\rho}$ fluctuates with each other, on the other hand, the SLA of the low $\bar{\rho}$ regions did not change with each other. The regions that

had the relatively low correlation coefficient agreed with the regions that had the extreme linear trends (see Fig. 2 and 4). We divided the SLA-KP into two regions according to the mean correlation coefficient; we roughly selected the threshold value as 0.5, which can separate the area having extreme trend and the remaining area. The MSLA of each region shows a good agreement each other (see Fig. 5). This demonstrates that the small-scale extreme features tend to cancel out and do not significantly impact MSLA-KP. This also suggests that the entire region can be treated as local variability fluctuating about

some background long-term mean, an important feature for this reconstruction procedure.

The purpose of the study of SLA-KP during the satellite era is to increase our understanding of SLA-KP before conducting the reconstruction of SLA-KP. To achieve this goal, an agreement between TG-KP and AVISO-KP was estimated in terms of correlation coefficient and linear trend by using averaged time series and individual time series at each TG location. These uneven patterns originated from two sources; one is river discharge in the Yellow Sea, and the other is a

vortex induced upwelling and downwelling effect in the East/Japan Sea area. The Dayang, Huli, Yingna, Zhuang, and Xiaosi Rivers flow into the Yellow Sea from China, and Yalu (Amnokgang), Taeryong, Taedong, Han, Geum, Mangyeong, Dongjin, and Yeongsan Rivers discharge into the Yellow Sea. The extreme patterns near the land seem to relate to the variation of river discharge. In the East/Japan Sea, both warm currents and cold currents exist simultaneously and the borderline repeatedly oscillates north and south. Near the borderline, the warm current and cold current make small gyres,

and the gyres make the uneven surface variations. These kinds of large variability sea level features make the assessments of the linear trend poor.

The linear trend at each TG location was estimated and it was compared with the nearest point in AVISO-KP (Fig. 6). The $\rho$ values between TG-KP and AVISO-KP were estimated and the mean $\bar{\rho}$ was about 0.72. The comparison showed that





only five TGs showed acceptable accuracies having less than 30% of difference with the AVISO-KP's linear trend. Eleven TGs showed more than 30% of underestimation and twenty-one TGs had more than 30% of over estimation. To figure out the effect of these disagreements, the MSLA-KP of AVISO was compared with the MSLA of TGs-KP, and these time series showed $\bar{\rho} = 0.89$ and NRMSE = 0.52 (see Fig. 7). The MSLA rise of combined TGs was estimated as 4.31 mm/yr and this

value is about 40% higher than the MSLA-KP of AVISO. This disagreement originated from the short time period and a lack of TGs. Unresolved vertical land motion at the TG-KP could also lead to such disagreements.

CSEOF decomposition was conducted to investigate periodic orthogonal behaviors for SLA-KP with 12 months nested period after removing mean values at each grid point. The first mode represented a seasonal variation considering the CSEOF's SLA patterns and the periodic PCT (Fig. 8). Nearly 60% of SLA-KP variations can be presented by this mode. The

second mode shows similar spatial patterns having positive value for all months, and the PCT shows clear positive trend (Fig. 9). This means the second mode is a sea level rising mode, and it represents 10% of variations of SLA-KP roughly. The third and fourth modes were not able to relate to specific phenomenon, and the modes explain variability in the SLA-KP at about 5% and 3%, respectively. Using the four modes, we can explain about 75% of SLA-KP. The first and second modes have the linear trend, but the linear trend in the first mode is negligibly small compared with the signal itself (Fig. 10).

Hence, we can say that the second mode is the most important key to estimating the sea level rise around the KP.

## 3.2 Sea Level Reconstruction around the KP

To begin the process of reconstructing sea level around KP, CSEOF decompositions (Kim et al., 2015; Kim and North, 1997; Kim et al., 1996) with twelve-months nested period were performed on both the AVISO- KP and the SST datasets as

described above. The datasets were decomposed into Loading Vectors (LVs) and corresponding time series of Principal Components (PCTs).

To reconstruct SLA-KP over 1900-2014, we then applied the multivariate regression accounting for lagged relationships, relying on CSEOF's modes of SST and AVISO-KP. For these reconstructions, two SST reanalysis datasets (ERSST and COBESST2) were used. Each SST data was divided into three cases: global, NWP, and the entirety of the KP region shown

in the figures. As a result, six cases of reconstructions were conducted and the six reconstructions showed a reasonable agreement with MSLA-TG over 1965-2014. For the period prior to 1965, however, the results showed considerable diversity (see Fig. 11). The mean reconstructed SLA-KPs (hereafter ReSLA-KPs) were compared with the mean reconstructed SLA of previous study (Hamlington et al, 2011; ReSLA-H) and the MSLA-TG from 1970-2009 considering the available number of TG data because there were a few TG data available before 1970. Both a correlation coefficient and normalized root mean

squared error (NRMSE) were applied for the quantified comparison. The comparison result is given in Fig. 12. Considering the NRMSE, we can see that the SST of NWP and KP provided better reconstructions than ReSLA-H because the NRMSEs of these cases are greater than ReSLA-H. However, considering the correlation coefficient, only SST of NWP datasets





showed better results than ReSLA-H. Finally, we selected the reconstruction using COBESST2-NWP (hereafter ReSLA-NWP) as the best reconstruction considering both NRMSE and correlation coefficient.

Most of reconstructions show better agreement than the reconstruction of Hamlington et al. (2011) in terms of correlation coefficients despite we did not use TG data during the reconstruction process. We compared MSLA-KPs from TG-KP, ReSLA-H, and the results of current study to check the reconstruction result. The mean ReSLA-KPs show good agreement with the mean ReSLA-H, but poor agreement with the MSLA-TG (see Fig. 13). This disagreement, however, is likely caused by lack of high-quality TGs before 1970. We further calculated the correlation coefficient, $\rho$, and linear trend using ReSLA-KP, ReSLA-H, and TG-KP. We made two correlation comparisons: one between ReSLA-KP and TG-KP, and the other between ReSLA-H and TG-KP to check if ReSLA-KP showed better representation of each TG-KP. The ReSLA-KP showed higher $\rho$ values than ReSLA-H (see Fig. 14a) demonstrating the better agreement between the current reconstruction and TG-KP. The linear trends of TG-KP, ReSLA-KP, and ReSLA-H were estimated at the TG location over 1970 to the present; for the calculation, each time series was edited to have the same time span data gaps. The estimated linear trends are given in Fig. 14b. Figure 14 indicates that the ReSLA-KP has similar linear trends with ReSLA-H at the TG location and the variance of the trends are smaller than TG-KP. ReSLA-KP comparing to ReSLA-H shows better agreement with the AVISO-KP over satellite era (see Fig. 15); it also has more fluctuations (see Fig. 13), which are important to apply this results for engineering purposes. These detailed fluctuations are closer to the actual sea level variability, and this is likely a result of the applied number of modes for the reconstruction process. Hamlington et al. (2011) used a limited number of CSEOF modes to avoid over-fitting issues, but in this study, nineteen CSEOF modes are used which explain 98% of total variance of SLA-KP.

Using MC simulation, a 95% confidence interval was estimated based on the best reconstruction case (COBESST2-NWP). By applying the regression coefficients' mean and standard deviation, each mode's PCT was randomly generated, and the process was repeated by thousand times and these PCTs were combined with CSLV's of AVISO-KP. Through this process, thousand of SLA-KP reconstructions were generated, and the mean and standard deviation were estimated using these. This means that the reconstructed data has their mean and standard deviation values. The resulting MSLA-KP and 95% confidence interval are shown in Fig. 16.

The linear trend in SLA-KP over 1900-2014 is estimated as 1.71 ± 0.04 mm/yr, and this value is similar to the linear trend of Church and White (2011) as 1.70 ± 0.02 mm/yr. A linear trend at each grid point of AVISO sea level anomaly data was calculated, and the maximum and minimum linear trends are about 2.1 mm/yr and 1.4 mm/yr, respectively (Fig. 17). The difference on the linear trends map of the reconstructed SLA-KP is much less than the AVISO-KP's linear trends over 1993-2015. This means that the long time period reduced the effect of large amplitude signals. This is particularly true for the high-trend areas in the Yellow Sea where trends were weakened significantly.





## 4 Summary

There were two primary goals of the work presented in this study: 1) Improve the understanding of the sea level around the KP both in the past and present and 2) Present a new reconstruction scheme for local areas with insufficient tide gauge coverage. To meet these goals, we used the satellite altimeter data from AVISO and the TG data from PSMSL to investigate the characteristics of SLA-KP. The linear trend of MSLA-KP was estimated as $3.1 \pm 0.5$ mm/yr from the satellite altimeter data (see Fig. 3). However, when we looked into the trend map, some areas (such as near the river mouth in the Yellow Sea and in the middle of the East/Japan Sea) showed significant departures from the mean (see Fig. 2). Understanding this spatial variability has important implications for future planning efforts around the KP.

To investigate this further, the reconstruction was performed using AVISO-KP and two SST reanalysis datasets. Each SST dataset was divided into three cases (global, North-west Pacific, around the Korean Peninsula). The six datasets were decomposed by CSEOF analysis; the AVISO-KP was decomposed into CSEOF modes after removing the GMSL. The decomposed CSEOF modes' CSLV played a role of basis functions for the reconstruction, and the main process of reconstruction was extending the PCTs of each mode into the past. The six reconstructed SLA-KPs were generated by this study over 1900-2014. Using the correlation coefficient and the normal root mean squared error, the best reconstruction was selected. The best reconstruction was produced by COBESST2 data of the North-west Pacific area. Through the best reconstruction results, the linear trend of SLA-KP was estimated as $1.71 \pm 0.04$ mm/yr. The extreme linear trends shown in Fig. 2 didn't appear in the reconstructed SLA-KP. This reconstruction showed better agreement than the previous study's result (Hamlington et al., 2011; see Fig. 16 and 17).

While we focus here on a specific example (the KP), this study can be used to inform other efforts in studying past, present and future sea level in areas with poor tide gauge coverage and significant future risks to impending sea level rise. Our interest was on the KP, specifically, but it was found that including information from the Northwest Pacific improved the localized representation of sea level. Consequently, considering large-scale ocean variability and teleconnections between different parts of the ocean is important when selecting the reconstruction domain. This study also demonstrates that tide gauges may not even be necessary to understand sea level in the past. Using only satellite-based sea level information and SST, we found dramatic improvements between the current reconstruction and past efforts, particularly when comparing to the tide gauge variability. Many tide gauges are influenced by vertical land motion that cannot easily be corrected for. Relying on SST alleviates concerns associated with non-ocean related trends. It should be noted that this reconstruction may not work as well in other parts of the ocean, especially those with a less pronounced agreement between sea level and SST. This study does, however, demonstrate the extended efforts that must be made to obtain accurate information about past sea level. As planning efforts get underway in more parts of the world, such comparisons between past and present sea level will become more important, and alternative approaches to simply using tide gauge information are going to be needed.





**Acknowledgement**

S.H.C. and K.D.S. were supported by Basic Science Research Program through the National Research Foundation of Korea (NRF) funded by the Ministry of Science, ICT and Future Planning (NRF-2014R1A2A2A01007921). B.D.H. acknowledges support from NASA PO NNX15AG45G and NNX16AH56G.

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



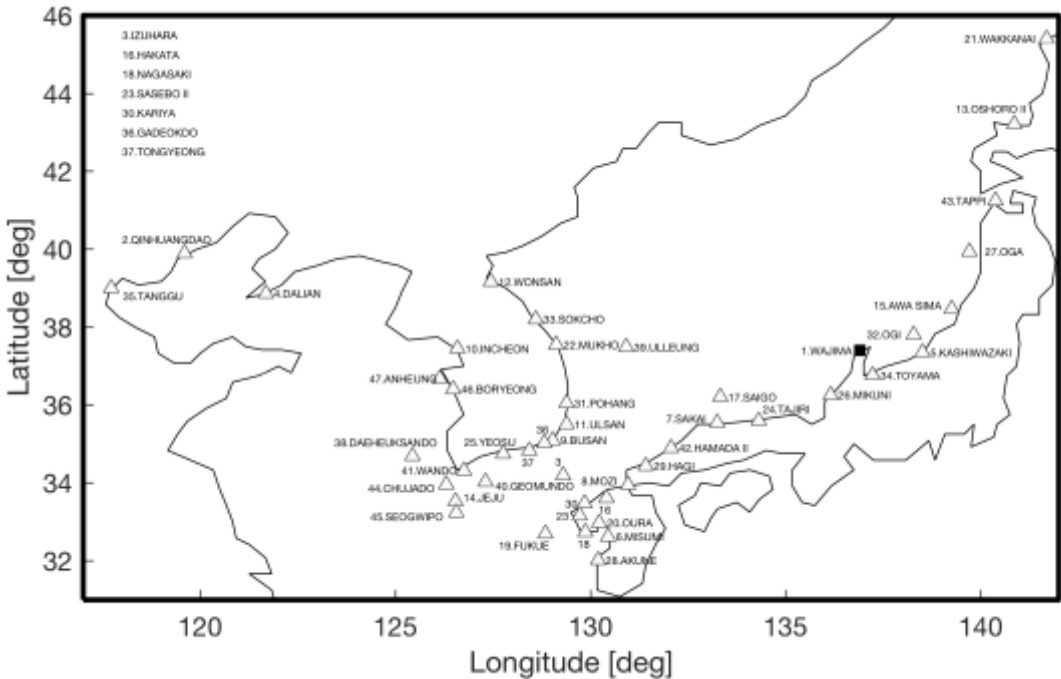

**Figure 1: TG locations around the Korean Peninsula. The black square is Wajima TG station which has the longest record length (1930-present)**

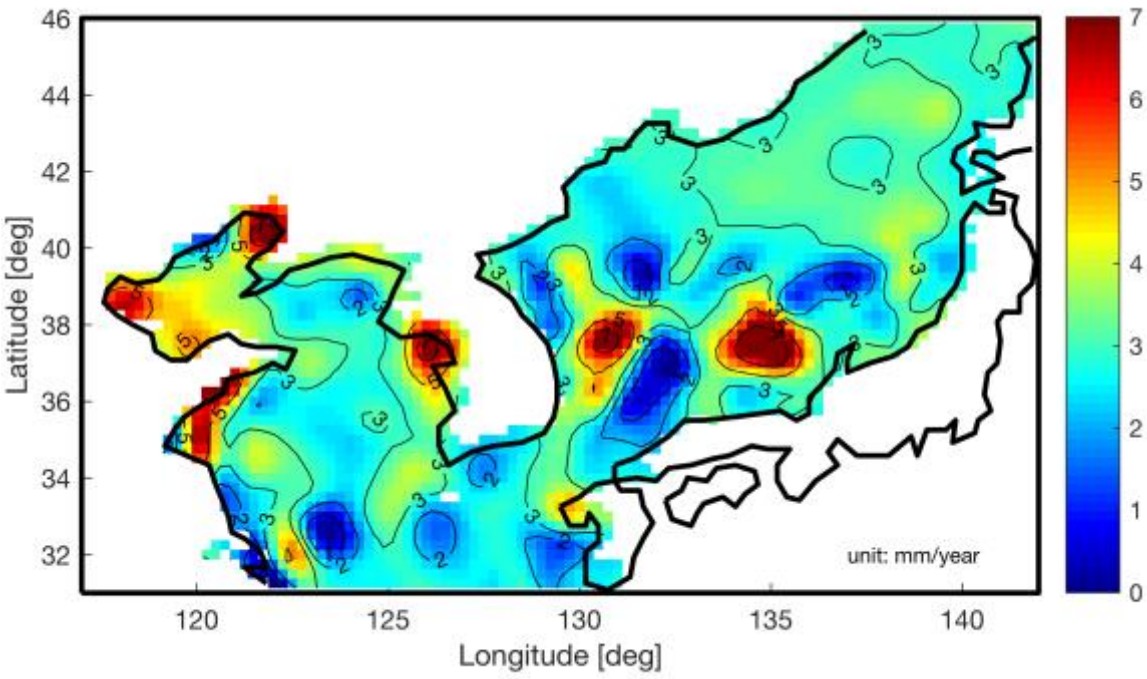



**Figure 2: Linear trend map of AVISO-KP (1993-2015)**

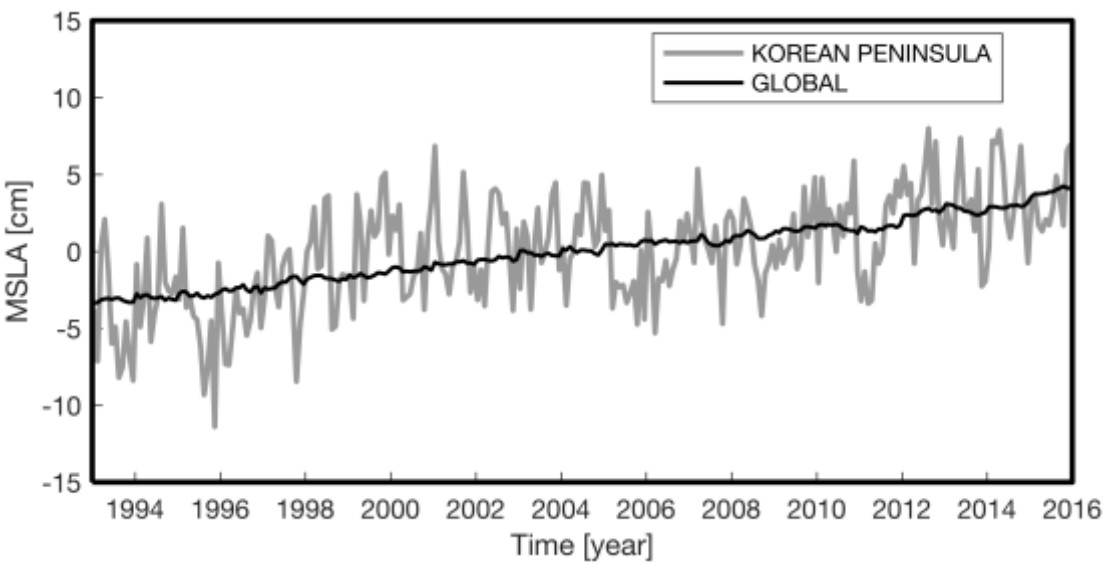

5   **Figure 3: Comparison between MSLA-KP and global MSLA time series (1993-2015)**

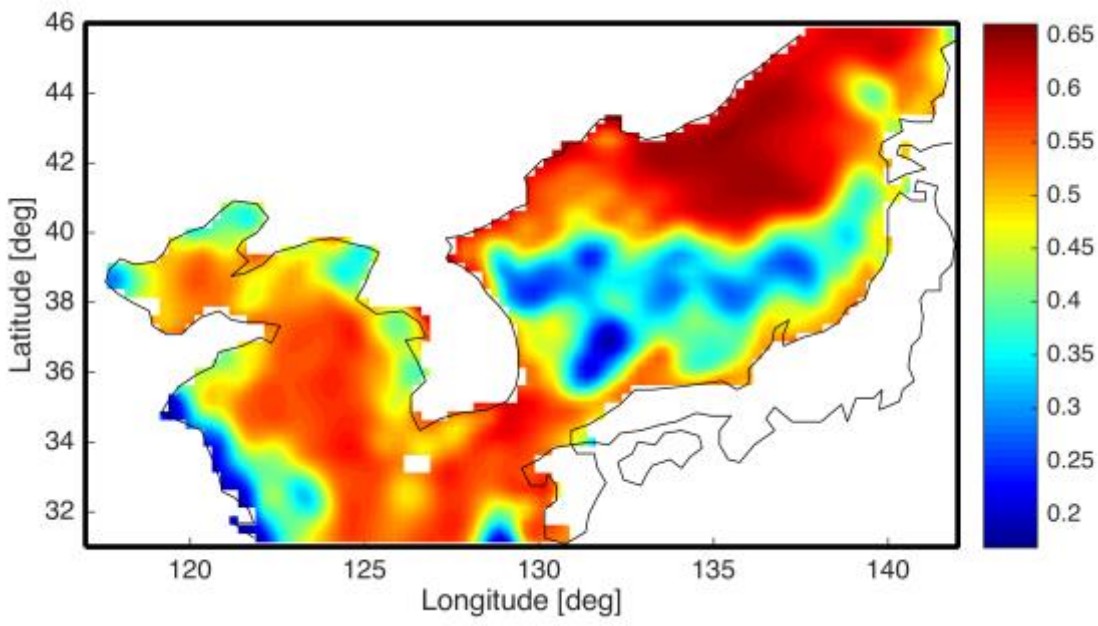

**Figure 4: Mean correlation coefficients between each grid's SLA and other grid's values (1993-2015)**



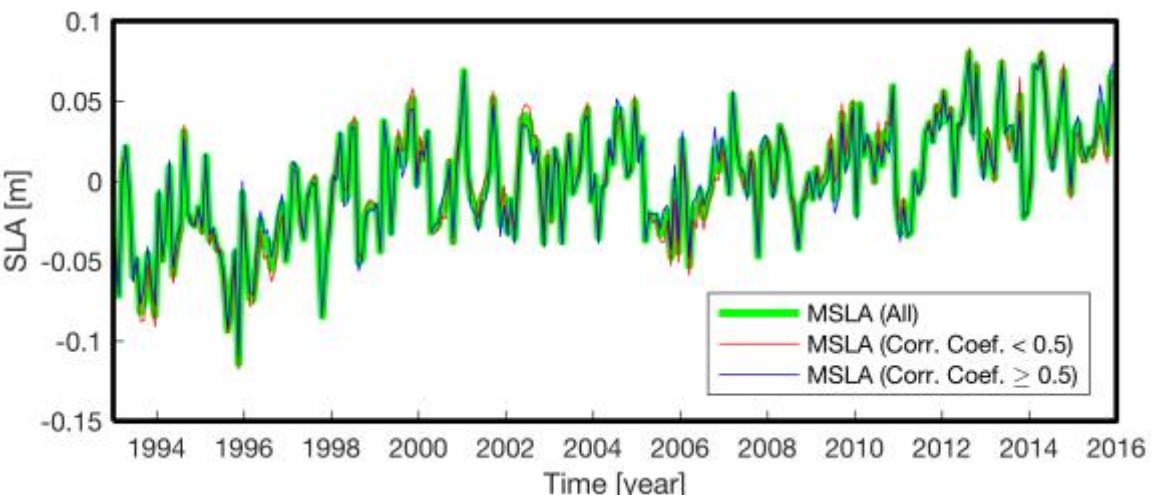

**Figure 5: Comparison of the mean SLA divided into two regions based on the correlation coefficients in Fig. 4**

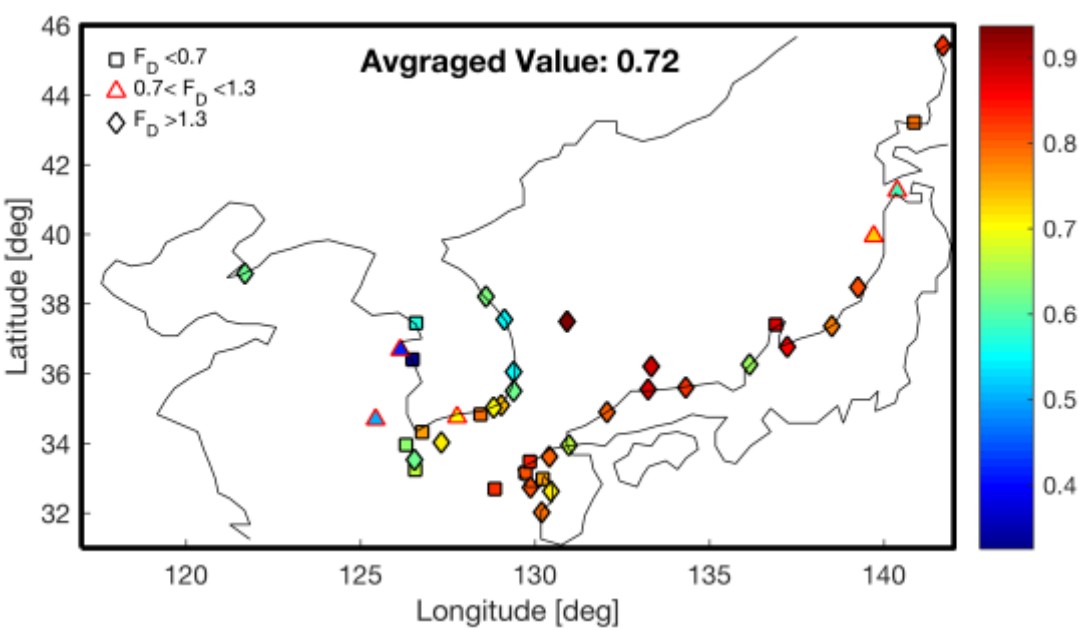

**Figure 6: Linear Trends comparison and correlation coefficients between TG-KP and AVISO-KP over 1993-2014, where $F_D = SLR_{TG}/SLR_{AVISO}$**



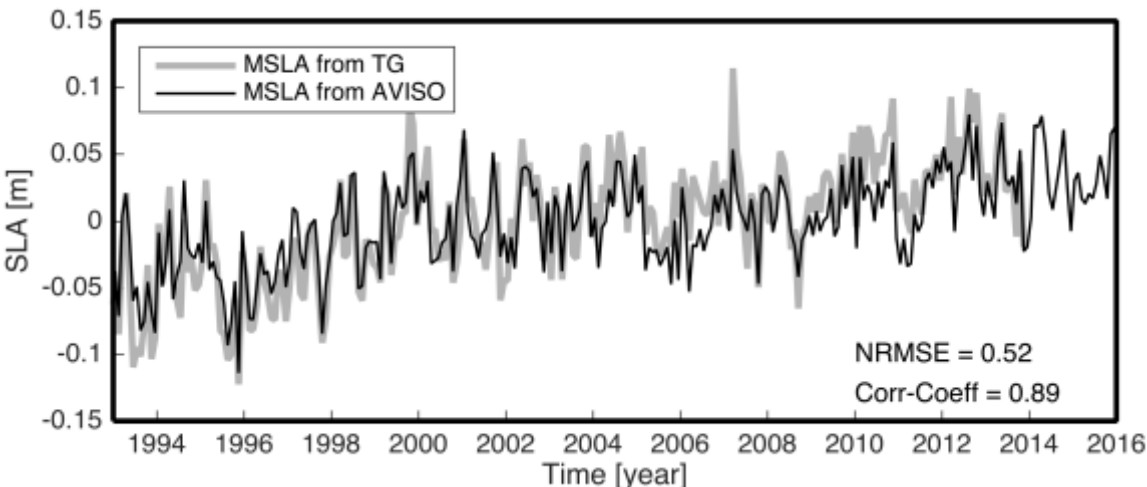

**Figure 7: MSLA-KP time series of AVISO and TG**

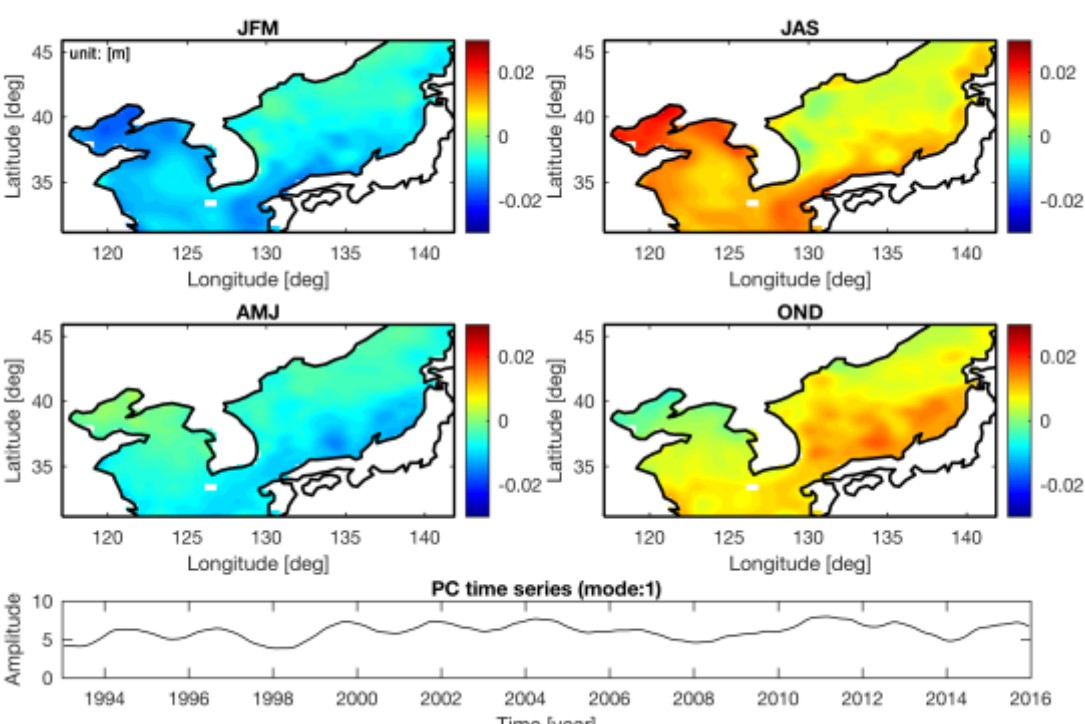

**Figure 8: The first CSEOF mode of AVISO-KP**





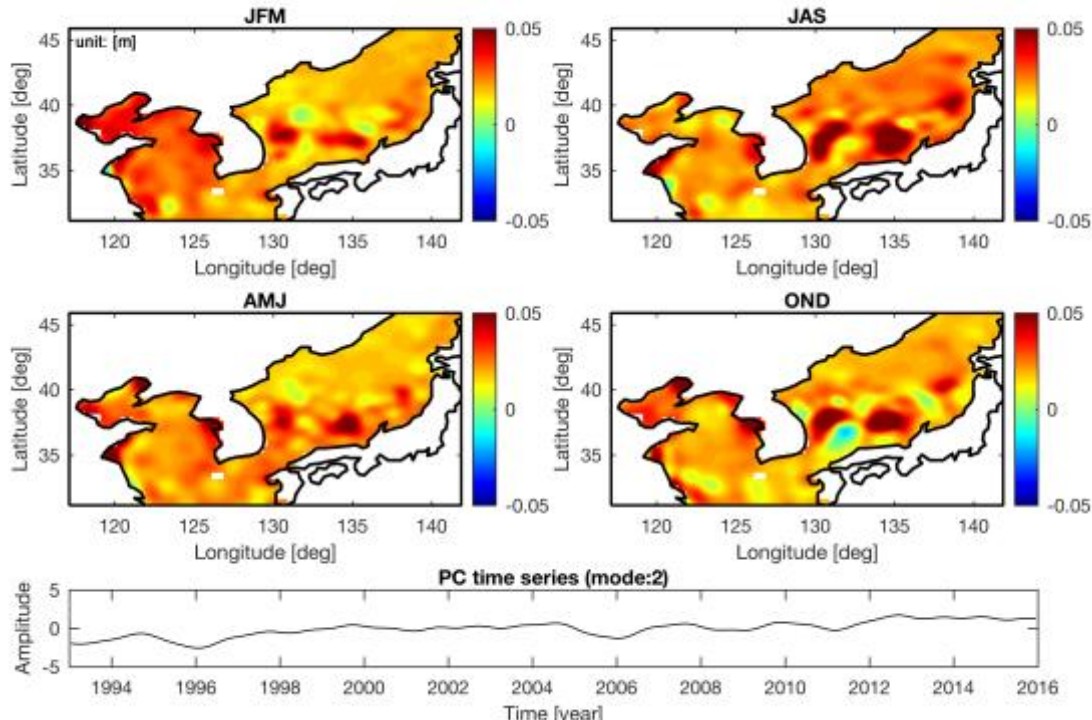

**Figure 9: The second CSEOF mode of AVISO-KP**

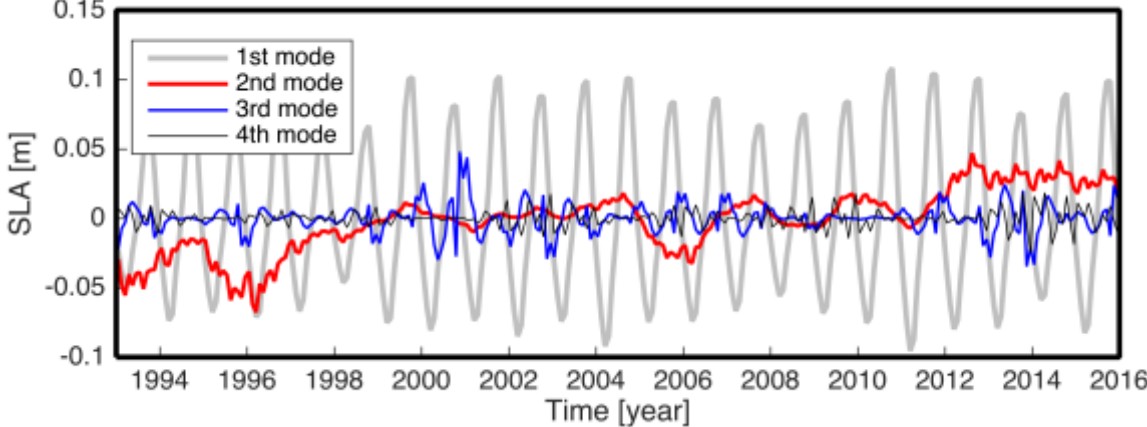

**Figure 10: MSLA of the four biggest modes of CSEOF decomposition**





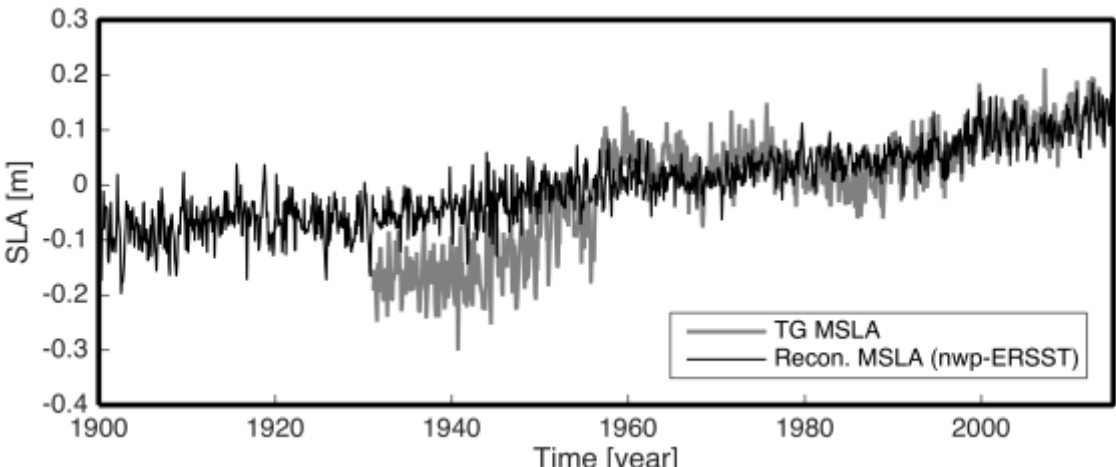

**Figure 11: Comparison between reconstructed MSLA and the TG MSLA (ERSST of the North-West Pacific)**

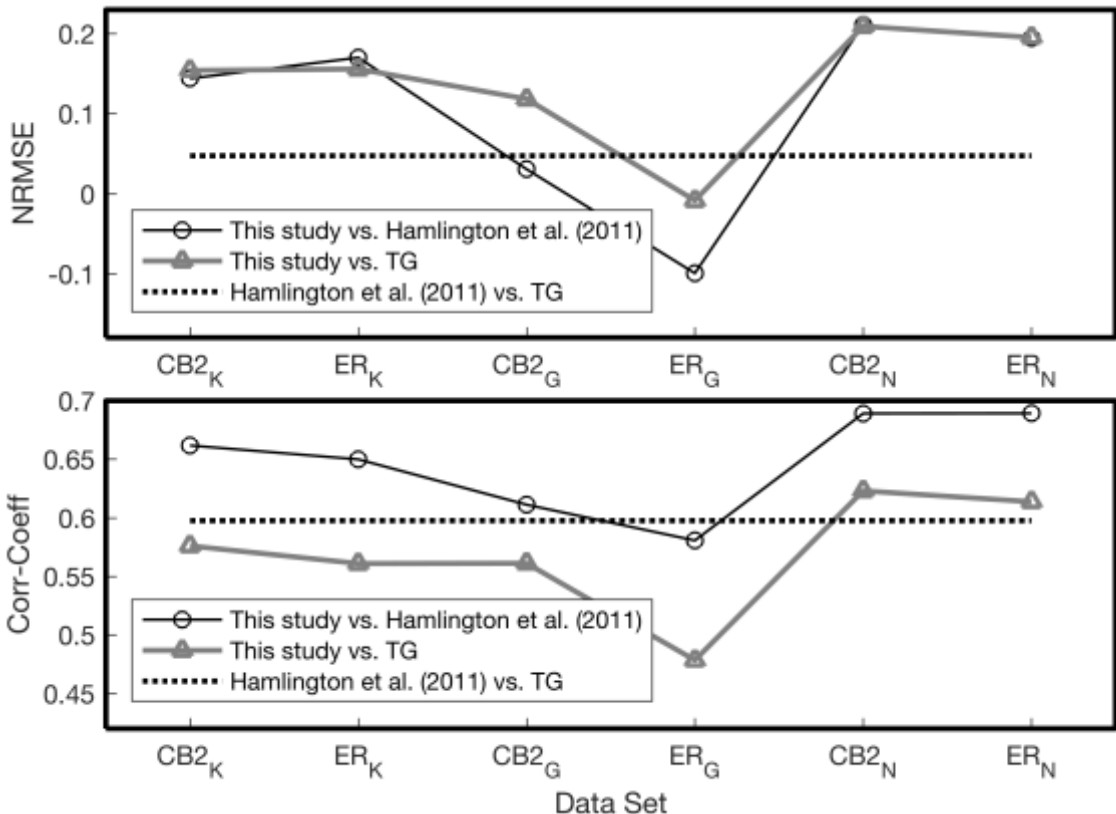

5   **Figure 12: Results of goodness of fit test for Reconstructed MSLA according to Hamlington et al. (2011) and TG MSLA; the top figure include normalized root mean squared error and the other include the correlation coefficients; here subscripts K, G, and N represent 'around the Korean Peninsula', 'Global', and 'the North-West Pacific', respectively and CB2 and ER represent 'COBESST2' and 'ERSST'.**





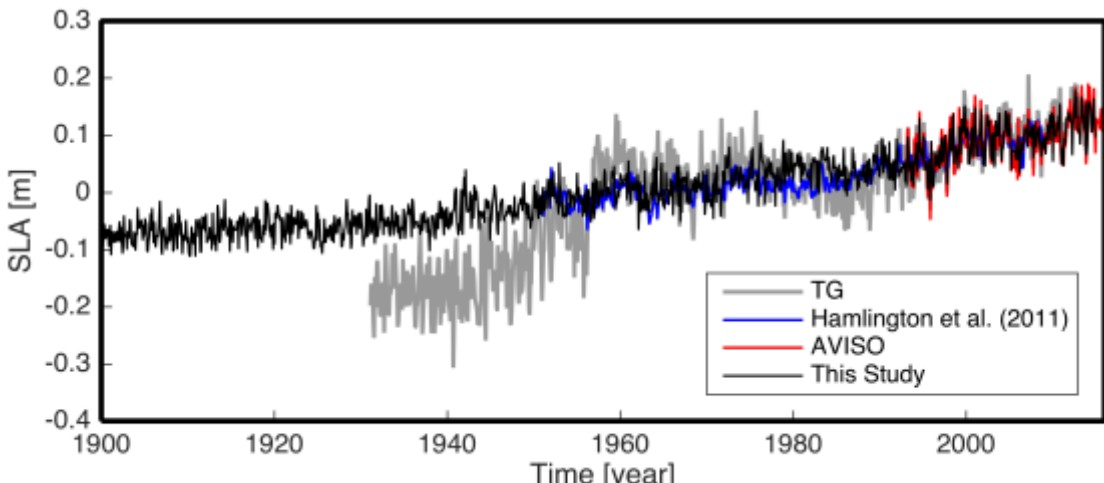

**Figure 13: Comparison of MSLA-KP time series**

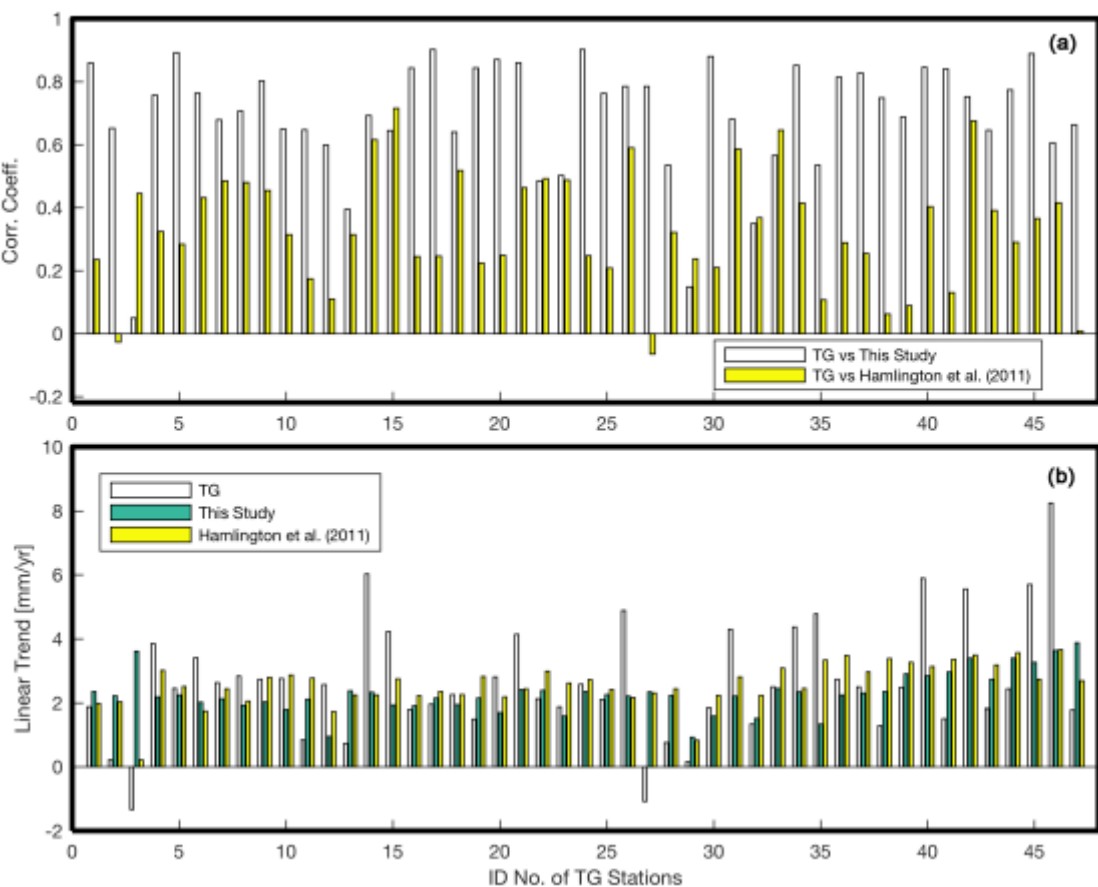

**Figure 14: (a) Comparison of correlation coefficients between TG-KP and the reconstruction results over 1993-2008; (b) Comparison of linear trends over 1970-2008**






**Figure 15: (a)Correlation coefficient map between Hamlington et al. (2011) and AVISO-KP over 1993-2008; (b) Correlation**
5  **coefficient map between this study and AVISO-KP over 1993-2008**




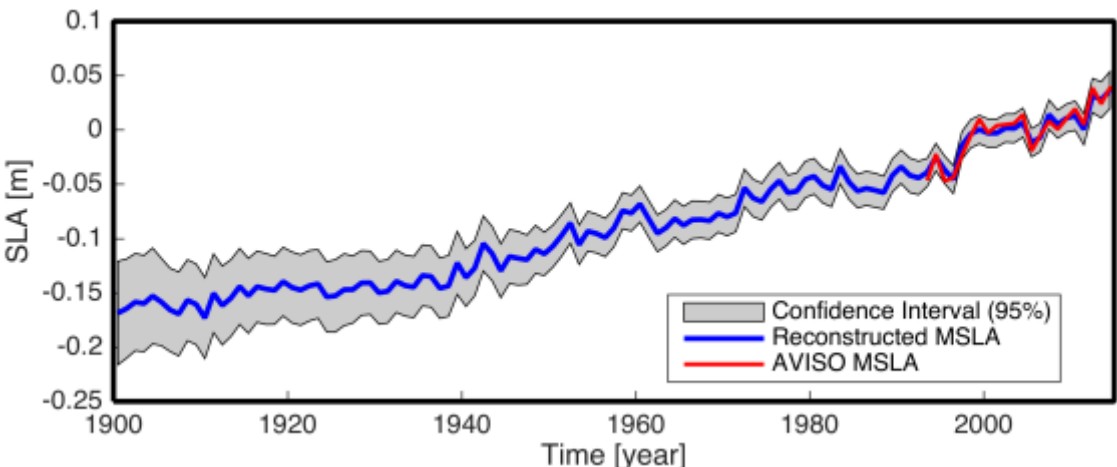

**Figure 16: The Best reconstructed MSLA (COBESST2 of the North-West Pacific Ocean) and 95% confidence interval.**

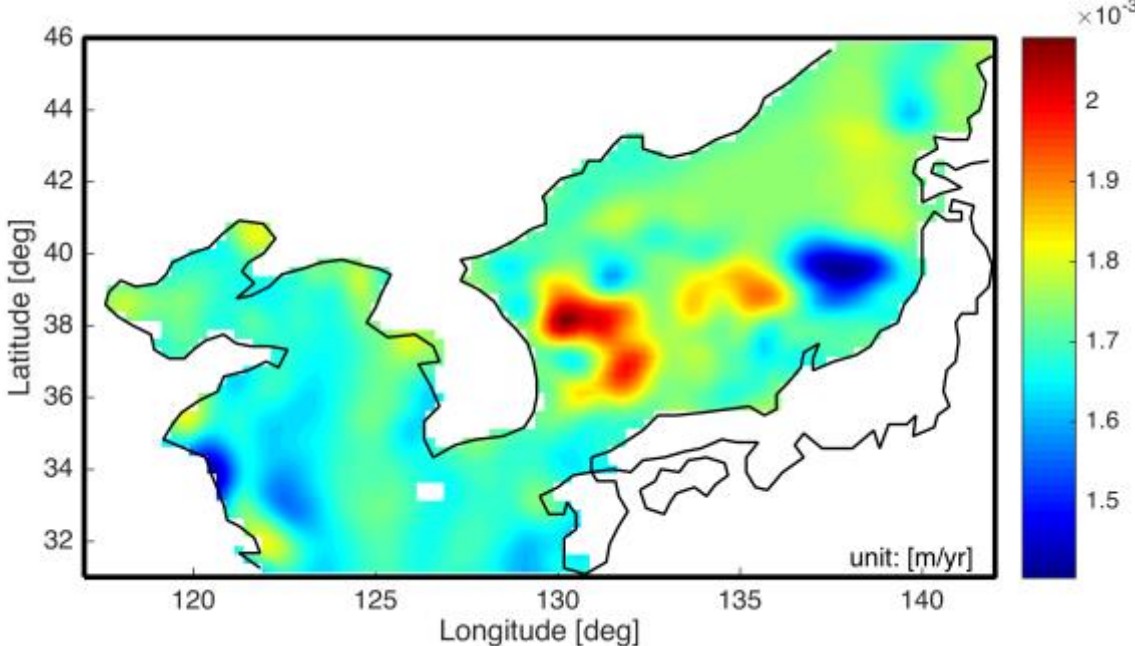

**Figure 17: Linear trend map of the reconstructed SLA-KP over 1900-2014**