# Peer review of "Reconstruction of Sea Level Around the Korean Peninsula Using Cyclostationary Empirical Orthogonal Functions"

_Ocean Science, 2017_

## Referee Comment (RC1) · Anonymous Referee #1 · 10 Jul 2017

The manuscript developed a new method in a regional sea level (SL) reconstruction using CSEOF and its multi-regression between SL and SST PCTs. The reconstruction is extended back to 1900 while sparse TG observations are not included. The reconstructed SL is better than a similar SL reconstruction in comparison with TG observations. The manuscript could contribute to the OS in understanding SL changes and SL reconstruction. I recommend the Editor accept the manuscript after a major revision.

Major comments (a) The sea level change may be associated with many factors such as ocean temperature (including SST), salinity, currents, and surface winds etc. There-

fore, the multi-regression between SST and SL PCTs may not include all aspects of SL changes. I am wondering whether the reconstruction could further be improved if more physical variables are considered. (b) The SL reconstruction does not include TG observations, but have a clear improvement over a similar reconstruction that includes TG observations. I am wondering whether the SL reconstruction could further be improved if all available TG observations are included. (c) How to validate the SL reconstruction in the early period over 1900-30 when no TG observations are available. It might be a little risky to include the reconstruction in this period. (d) Writing and presentation may need improving. There are too many abbreviations such as SL, MSL, GMSL, SL-KP. For example, MSL and GMSL could be explained in figure captions. KP is unnecessary because the study focuses on KP region only. Figure captions should identify the data source and average region etc.

Detailed comments (P=page, L=line) P1L11, revise: extend the spatial resolution .. into the past P3L5, CSEOF is not defined P3L14, "KP" could be deleted throughout the manuscript since the study has been limited over the KP region anyway, which will greatly improve the readability. "KP" could be noted in the figure caption when necessary. P3L21, revise: looking at the regional level will lead to P4L11, annual signal => seasonal signal? P4L20, include data => included data? P4L21, over => from? P6L5-7, revise the sentence P6L11, delete "in this case", "really" P6L12, independent of => independent from? P9L3, How does "summing" actually do, arithmetic or square-root? P10L10, this is an indication that SL is not merely dependent on SST. P11L22, delete "then" P11L25, delete "cases of" P11L28-29, delete "considering the available number of TG data" P12L4, It is not clear how MSLA-KP is defined (assuming every ocean grid in reconstruction). How MSLA-KP can be compared with TG-KP (only in TG grids). P12L12, revise "was edited to have the same time span data gaps" P12L14-15, revise the sentence: ReSLA-KP show a better agreement of AVISO-KP than ReSLA-H. P12L17-18, how many modes are used in Hamlington? P12L23, thousand => a thousand P13L17-18, authors should extend the conclusion of a better current SL reconstruction. there is no way from Figures 16-17 to tell the current study is better. It is

not clear in Figure 13 either. It may be necessary to point to Figure 14a. A better way is to calculate the RMSE.

Fig. 1, digital quality should be improved. Fig. 2, coastal line should be consistent with those in other figures. Fig. 3, I could caption the figure as "Mean SLA in KP (gray) and global (black) regions from AVISO" so that I can get rid of some abbreviations. Fig. 4, add "AVISO" in caption Fig. 5, add "AVISO" in caption Fig. 6, revise: trends (shapes) and correlation (color), change the red color of triangle into black so that the color will not be confused with correlation. Fig. 7, NRMSE, I don't know the advantage of using normalized RMSE instead of RMSE. Figs. 8-9, I am confused how the 3-month averaged mode is plotted. I assume there is only one CSEOF associated with one PCT for a particular mode. Fig. 10, I assume this is for KP region Fig. 11, which region, KP region? Fig. 12, Why does Hamlington have a constant Corr and NRMSE? Fig. 14, "yellow" is barely identifiable. Why the correlation is over 1993-2008 while trend is over 1970-2008? Fig. 15, The figure look great but there is a question: Since the study uses the CSEOF derived from AVISO, therefore validation against AVISO is considered to be not independent. One may argue that if authors use Hamlington deriving CSEOF, the performance reconstruction may be close to Hamlington.

---

## Referee Comment (RC2) · Anonymous Referee #2 · 15 Aug 2017

This MS is trying to present a sea level reconstruction method targeting on the marginal seas around Korean Peninsula. Understanding the regional variations and therefore attempting to reconstruct/project are interesting research areas in sea level science. Unfortunately, there are many key issues in this version of ms. The paper is tedious to read with loosen focus, and in some places it is hard to understand. Ultimately, I think the presentation is needed to improve. A major revision is suggested before getting this paper published in OS.

Some key questions are provided first, followed with some minor comments.

Title and abstract: sea level projections seem to be an important aspect in this paper.

[Figure]

Unfortunately, there are no results related to 'projections'.

Introduction: authors believe that the sea level reconstruction using SST provides better results than the conventional methods (using TGs). However, SST was also sparsely observed in early years including ICOADS. How well do the SST methods cope with this common concern? Clarification is needed. I still believe the wind stress and local surface currents are dynamically important for sea level variations, like many studies have shown. There is no direct link between coastal sea level and SST in open ocean. How possible to include other dynamical factors?

Section 2: this part reads loosen and tediously long, and many parts are unnecessarily mentioned with many times. I would suggest shortening this section with concise contents to avoid readers losing interests.

Section 3: This section again is not properly presented. Essential questions: 3.1: I do not think the following key question is answered. 'To reason whether the extreme trends patterns was related to the local mass distribution caused by various sources such as vortex and river discharge or was an independent....' The extreme trends on China coasts in Fig 2 are proposed as a result of increasing river discharge by the authors. However, there is no convincing evidence supporting this. (one would not expect that river discharge can cause sea level increase on north Chinese coasts, because it is drying over recent years in this region). Same for the ocean current impacts. Can authors provide evidences supporting this (P10 Lines 28-31)?

Also, I cannot see any point of separating the regions with local correlations </>0.5. Because the two regions are both located in Yellow Sea and Japan Sea, the regional averages are supposed to not contain local information, and they instead reflect the large-scale variations. This might be reason why the two series in Fig5 are always highly correlated.

For the correlation map e.g. Fig4 (and Fig 6), is the annual cycle removed? Removing the seasonal cycle is critical. Otherwise, they are always statistically correlated but it

does not make any sense. Need to clarify.

How can the sea level records between TG and AVISO be correlated e.g. Fig 6 when also having linear trends? If linear trends exist, they are always correlated. Correlation is for assessing the similarity between detrended variability/anomalies but cannot be used for assessing the trends. The basic concept I think is wrong. Please clarify.

Fig 6 & 7: how far are the AVISO sites from TG stations? Fig8 & 9: I cannot see there is a trend in the PC series of Fig9. What are the trend value and its significance level? Does it agree with the values based on the local estimation i.e. Fig 3. Because there is no annual cycle signal in Fig9, there is no need of presenting it with 4 seasons.

Section 3.2: what are the reasons for COBESST2-NWP having best correlations with sea level? Do author have interpretations? Why does not the local SST do better job than others? Also, the short names e.g ReSLA-NWP are not used in figures, which however use the long name. Authors need to be careful for the presentation throughout the whole paper.

Again in Fig 14 & 15, are the linear trend and annual cycle both removed before calculating correlation? Are the trends in Fig 14b statistically significant? 'these detailed fluctuations are closer to the actual sea level variability': what is the actually sea level variability?

Authors seem to insist that the SST-based reconstruction shows better results. What are the reasons for that? In the marginal seas of NWP, many studies have shown that the local ocean surface currents and wind tress determine the sea level, and the open ocean in far-field has less impacts. However, this paper finds the (far-field) NWP SST can 'statistically' better capture the sea level in marginal seas of NWP i.e. KP. What is the science behind it? Please keep in mind that the sea level variations between the two sides of western boundary currents (Kuroshio/Oyashio) are very differently forced e.g. by the thermalsteric height and open ocean currents via geostrophic balance and by local wind/surface currents.

[Figure]

More essentially, this paper is focusing on 'reconstruction capability', but it spent a lot space in section 3.1 comparing TG and AVISO. Authors should work properly to make the presentation and structure of this paper concise and focused.

Conclusion: What is the linear trend map of reconstructed SLA-KP over satellite era? Are they comparable with Fig 2? How are the SST variations looking like over this region/NWP? Does SST follow the sea level changes very well?

Minor comments. For example: P2, lines 6-7: do not understand. What does bias mean? Also, references are needed to support this statement. P2, lines 29-31: references? P2 lines: 31-32: do not understand. P2 line 33: this needs to reword P3 lines 14-15: what reconstructions? P3 lines 11-22: references? P3 lines 11-27: the focus/motivations are loosen and not concise. P4 lines 9-10: do not understand Figure 1 is not readable Figure 1 seems to have 3 TGs on China coasts, while there is only one appearing in Figure 3. Any flags applied?

---

## Author Comment (AC1) · 13 Sep 2017

**Response to the major comments**

We thank the reviewer for the comments and insight into the paper. We have made many adjustments to the paper and have added clarifications where necessary. We think the paper is much improved as a result.

(a) The sea level change may be associated with many factors such as ocean temperature (including SST), salinity, currents, and surface winds etc. Therefore, the multi-regression between SST and SL PCTs may not include all aspects of SL changes. I am wondering whether the reconstruction could further be improved if more physical variables are considered.

To apply multi-variables to current reconstruction scheme, there are several problems. First, when we applied multi-variable's PCTs as predictor, the over-fitting occurred because as the mode goes higher, the possibility of overfitting increases. Second, the reanalysis of other variables does not appear to be of sufficiently high quality back through time, based on our tests. Actually, we tried to reconstruct SLA using wind and SST data simultaneously, but the result is poorer than each individual reconstruction case. To solve these problems, we it will require significant effort beyond the scope of this first study with the goal of demonstrating the basic technique. However, we do agree the reconstruction applying multi-variables is valuable topic for the future study.

(b) The SL reconstruction does not include TG observations, but have a clear improvement over a similar reconstruction that includes TG observations. I am wondering whether the SL reconstruction could further be improved if all available TG observations are included.

To include TG data, there are two problems. First, using TG data we cannot conduct CSEOF analysis because they have lots of discontinuous points and their spatial coverage are too poor. So we cannot establish the proper regression relationship between TG and SST. Second, the TG data's quality is not good. The vertical land motions cannot be explained and accounted for and many of the Japanese TG are impacted by earthquakes or volcanic activities. To use TG data, the additional researches are necessary to correct the vertical land motions.

(c) How to validate the SL reconstruction in the early period over 1900-30 when no TG observations are available. It might be a little risky to include the reconstruction in this period.

This is indeed an issue with any reconstruction study. As we can see in Fig. 11, even we have TG data for vilification, prior to 1970, the agreement between MSLA from TG and ReSLA-KP is very poor. But we thought that it does not mean our reconstruction is not good because the TG data is not enough to verify. Even we cannot verify the reconstruction results, we think the result is still valuable. However, we have tried to make clear in the paper that the quality of the reconstruction is difficult to assess into the past.

(d) Writing and presentation may need improving. There are too many abbreviations such as SL, MSL, GMSL, SL-KP. For example, MSL and GMSL could be explained in figure captions. KP is unnecessary because the study focuses on KP region only.

We have modified the abbreviations. We have, however, left in the KP in some locations, to distinguish from the global terms.

(e) Figure captions should identify the data source and average region etc.

We have made this change to the figure captions.

**Detailed comments**

P1L11, revise: extend the spatial resolution ..into the past

| Original | Many studies have been conducted to extend the spatial resolution of the satellite data into the past by finding novel ways to combine the satellite data and tide gauge data in what are known as sea level reconstructions. |
| --- | --- |
| Revised | Many studies have been conducted to create a dataset with the spatial coverage of the satellite datasets and temporal length of the tide gauge records by finding novel ways to combine the satellite data and tide gauge data in what is known as a sea level reconstruction. |

P3L5, CSEOF is not defined

| We have now defined CSEOF in P2L22. |
| --- |

P3L14, "KP" could be deleted throughout the manuscript since the study has been limited over the KP region anyway, which will greatly improve the readability. "KP" could be noted in the figure caption when necessary.

| We have deleted 'KP' and add explanation about the default domain. |
| --- |

P3L21, revise: looking at the regional level will lead to

| Original | As mentioned above, TG coverage is poor extending back into the 20th century, and looking at the regional level will lead to relatively few gauges to analyse in most areas. |
| --- | --- |
| Revised | As mentioned above, TG-KP coverage is poor extending back into the 20th century, with many areas completely lacking tide gauge records. |

P4L11, annual signal=> seasonal signal?
The terms, annual signal and seasonal signal, are having same meaning. But to prevent confusions, we change seasonal signal to annual signal.

P4L20, include data => included data?
We have fixed it.

P4L21, over => from?
We have fixed it.

P6L5-7, revise the sentence

| Original | If previous reconstruction schemes are applied that rely only on sea level, then it is likely only possible to obtain reliable results after 1970. |
| --- | --- |
| Revised | If a reconstruction scheme for SLA-KP relies only on TG data, then the results are only reliable after 1970 when TG coverage improves. |

P6L11, delete "in this case", "really" P6L12, independent of => independent from?
We have erased this sentence.

P9L3, How does "summing" actually do, arithmetic or squareroot?
We mean Root Sum Square. This part has been deleted.

P10L10, this is an indication that SL is not merely dependent on SST.
We have deleted this part. And we gave up to explain the physical reasons for the extreme SLR values.

P11L22, delete "then"
We have deleted it.

P11L25, delete "cases of"
We have deleted them.

P11L28-29, delete "considering the available number of TG data"
We have deleted them.

P12L4, It is not clear how MSLA-KP is defined (assuming every ocean grid in reconstruction). How MSLA-KP can be compared with TG-KP (only in TG grids).
We have added more explanations as follow.

To check the reconstruction results, we calculated MSLA of TG-KP, ReSLA-H, and ReSLA-KP. Spatial mean was calculated for the two grid datasets. For TG-KPs, we calculated mean differences between each time steps and we integrated the differences. The integrated mean differences became the MSLA of TG-KP.

P12L12, revise "was edited to have the same time span data gaps"
We have erased 'data gaps'.

P12L14-15, revise the sentence: ReSLA-KP show a better agreement of AVISO-KP than ReSLAH.
We have revised.

P12L17-18, how many modes are used in Hamlington?
We have added detailed number.
Hamlington et al. (2011) used a limited number (< 90% of total variance) of CSEOF modes to avoid over-fitting issues, but in this study, nineteen CSEOF modes are used which explain 98% of total variance of SLA-KP.

P12L23, thousand => a thousand
We have corrected.

P13L17-18, authors should extend the conclusion of a better current SL reconstruction. there is no way from Figures 16-17 to tell the current study is better.
We have deleted this part.

It is not clear in Figure 13 either. It may be necessary to point to Figure 14a. A better way is to calculate the RMSE.
We have added more figures.

Fig. 1, digital quality should be improved.
I think it has a high resolution, 600 ppi. We will ensure that it is of sufficiently high resolution in any future submission.

Fig. 2, coastal line should be consistent with those in other figures.
We have changed the figure.

 Fig. 3, I could caption the figure as "Mean SLA in KP (gray) and global (black) regions from AVISO" so that I can get rid of some abbreviations.
We have changed the caption and figure.

Fig. 4, add "AVISO" in caption
We have modified.

Fig. 5, add "AVISO" in caption
We have modified caption.

Fig. 6, revise: trends (shapes) and correlation (color), change the red color of triangle into black so that the color will not be confused with correlation.
We have modified the caption and figure.

Fig. 7, NRMSE, I don't know the advantage of using normalized RMSE instead of RMSE.
'NRMSE' and 'RMSE' very similar, but when NRMSE has 'zero' value this means the regression is same with some constant value cases and if the value are negative that the compared data is less agreed its mean value. So, we believe that NRMSE gives some intuitive interpretation.

Figs. 8-9, I am confused how the 3-month averaged mode is plotted. I assume there is only one CSEOF associated with one PCT for a particular mode.
Yes, you are right. But the evolution is small through the months, therefore we represent the results as seasonal mean values to save some space.

Fig. 10, I assume this is for KP region
We have modified the caption.

Fig. 11, which region, KP region?
We have modified the caption.

Fig. 12, Why does Hamlington have a constant Corr and NRMSE?

Because we have 6 cases but ReSLA-H is just one case.

 Fig. 14, "yellow" is barely identifiable. Why the correlation is over 1993-2008 while trend is over 1970-2008?
I think the color problem is related to the resolution, I provide 600 ppi image and that figure has no problem to recognize yellows.
And the time period is 1970-2008 for the both cases.

Fig. 15, The figure look great but there is a question: Since the study uses the CSEOF derived from AVISO, therefore validation against AVISO is considered to be not independent. One may argue that if authors use Hamlington deriving CSEOF, the performance reconstruction may be close to Hamlington.

Yes, that's right. Nevertheless, ReSLA-H has very poor agreement. We just want to show the limit of global reconstruction, as you can see the below figure. Over 1993-2015, the correlation coefficient is pretty high, this means if we applied Hamlington et al. (2011)'s method in local scale, the correlation coefficients must be higher than current Fig.

---

## Author Comment (AC2) · 13 Sep 2017

**Response to the major comments**

We thank the reviewer for the comments and insight into the paper. We have made many adjustments to the paper and have added clarifications where necessary. We think the paper is much improved as a result.

1. Title and abstract: sea level projections seem to be an important aspect in this paper. Unfortunately, there are no results related to 'projections'.
The reviewer is correct – we have removed 'projection' from the title.

2. Introduction: authors believe that the sea level reconstruction using SST provides better results than the conventional methods (using TGs). However, SST was also sparsely observed in early years including ICOADS. How well do the SST methods cope with this common concern? Clarification is needed.
We do not necessarily believe that a reconstruction of SST will yield better results than a reconstruction using tide gauges. If we had a large number of high quality (long) TG records, the reconstruction using TG data would likely be the best. Unfortunately, TG data around the KP is less than 10 prior to mid 1960s. As a result, we have used SST data instead of TG data for the basis of our reconstruction. The question the reviewer raises is a difficult one to address in any reconstruction attempt – specifically, how accurate is the reconstruction in the past. To address this question, we have used several different cases, using different datasets and different areas of study. Through these test cases, we attempt to validate the reconstruction back through time, but certainly expect the reconstruction to be of lower quality towards the beginning of the record. We have tried to make this point clearer throughout the paper.

I still believe the wind stress and local surface currents are dynamically important for sea level variations, like many studies have shown. There is no direct link between coastal sea level and SST in open ocean. How possible to include other dynamical factors?

It is true that other variables could be used to reconstruct sea level. We have relied on SST in this case due to the availability of a long record and reasonably consistent measurements back through time. It should also be noted that we are not requiring a strong physical relationship between SST and sea level. Instead, we are requiring that there is a strong statistical relationship between the CSEOF modes of the two variables. This strong relationship has been demonstrated when we analyze both the SST and sea level data through CSEOF analysis (Hamlington et. al., 2011, 2012a). Actually, many CSEOF modes shows great agreement, e.g., ENSO and PDO (see, below figures. LHS and RHS are ENSO modes of AVISO and OISST, respectively; c.f. we just introduce these to show the example of their relationship)

[Figure]

As explained in 2.2.1 and 2.2.2, we tried to find the lagged relationship between PCts (SST and sea level), and indeed we could likely only physically interpret a few modes. Even if we cannot explain the exact physical background of each modes and their relationship in SST and sea level, the reconstruction results are still valuable. On a basic level, the only goal is to reproduce the PCts back through time. As long as the statistical relationship we have established between SST and sea level holds, the two can be used in tandem to reconstruction sea level.

We have attempted a reconstruction case using wind data in the place of SST, but the result is not good when compared to SST (see below figure). Other variables could be used, but it is beyond the scope of the current paper.

[Figure]

3. Section 2: this part reads loosen and tediously long, and many parts are unnecessarily mentioned with many times. I would suggest shortening this section with concise contents to avoid readers losing interests. We have trimmed this section and removed repeating text.

4. Section 3: This section again is not properly presented.
We have modified this section.

4.1 Essential questions: 3.1: I do not think the following key question is answered. 'To reason whether the extreme trends patterns was related to the local mass distribution caused by various sources such as vortex and river discharge or was an independent. . ..' The extreme trends on China coasts in Fig 2 are proposed as a result of increasing river discharge by the authors. However, there is no convincing evidence supporting this. (one would not expect that river discharge can cause sea level increase on north Chinese coasts, because it is drying over recent years in this region). Same for the ocean current impacts. Can authors provide evidences supporting this (P10 Lines 28-31)?
We agree with your comment. There are no studies that explain the relation between the sea level rise trend and ocean current (or river discharge). To explain the relationship, we need more research that is beyond this paper's boundary. This was largely speculative and as a result, we have simply decided to remove this part of the paper.

Also, I cannot see any point of separating the regions with local correlations </>0.5. Because the two regions are both located in Yellow Sea and Japan Sea, the regional averages are supposed to not contain local information, and they instead reflect the large-scale variations. This might be reason why the two series in Fig5 are always highly correlated.
This study's basis assumption is the SLA-KP can be represent with GMSL as the baseline and our reconstruction as the variability about that long-term trend. And we wanted to know the influence of the extreme trend zones to the mean sea level anomalies in KP. If the extreme zones had significantly different MSLA with the other zone showing normal SLR trends, we needed to conduct reconstructions separately, as shown below figure.

[Figure]

So, to separate out the extreme trend zone, we estimated the averaged correlation coefficient maps. As the reviewer said if the large-scale variations reflect the two separated zones' MSLA. That means we don't need to worry about the different trend zones.

In summary, we conducted these process to estimate the extreme trend zones influence to the MSLA-KP.

For the correlation map e.g. Fig4 (and Fig 6), is the annual cycle removed? Removing the seasonal cycle is critical. Otherwise, they are always statistically correlated but it does not make any sense. Need to clarify.
Yes, we removed the annual cycle. We have clarified it on the figure captions.

How can the sea level records between TG and AVISO be correlated e.g. Fig 6 when also having linear trends? If linear trends exist, they are always correlated. Correlation is for assessing the similarity between detrended variability/anomalies but cannot be used for assessing the trends. The basic concept I think is wrong.

We removed linear trends during the calculation of correlation coefficient. In this case, however, the linear trend represents very small variance compared to the fluctuations, so the trend has litte effect on the correlation coefficient values (see below figure). Regardless, the trend has been removed.

[Figure]

The trends and correlation coefficients in Fig. 6, actually, calculated separately, there were several figures before. To reduce the figures, we combined the information and put the information in one figure. See the below figures.

[Figure]

Figure 3. SLA linear trends over 1993-2013 using TGs_KP

Figure 4. SLA linear trends over 1993-2013 using AVISO_sla

Figure 6. Correlation Coefficients between TGs_KP and AVISO_sla points with seasonal signals

Please clarify. Fig 6 & 7: how far are the AVISO sites from TG stations?
I clarified the maximum distance (about 12 km) at the caption.
In Fig. 7, but, to calculate MSLA of AVISO, I just used entire area not the closest point.

Fig8 & 9: I cannot see there is a trend in the PC series of Fig9.
We have changed the Fig.8 and 9 to show bigger PCT. We think the trend is now apparent.

What are the trend value and its significance level?

To help readers' understanding, we have added one more figure. Fig 12 shows the linear trend values of each mode and their confidence intervals.

Does it agree with the values based on the local estimation i.e. Fig 3.
Fig 3 doesn't have annual signal. If you see Fig 11, the wiggled signals arise from the summation of each CSEOF modes. So, it does agree with MSLA-KP in terms of low-frequency signal.

Because there is no annual cycle signal in Fig9, there is no need of presenting it with 4 seasons.
The spatial pattern of one CSEOF analysis is not a single map, so they need to represent through their nested period. Actually, most of the CSEOF mode do not have similar spatial patterns though the time evolution. The reason why we can determine the 2$^{nd}$ mode as the trend mode, is these spatial patterns are pretty similar though the nested period. In other words, the non-periodic nature of the trend is represented by the CSEOFs as a spatially constant map.

4. 2 Section 3.2: what are the reasons for COBESST2-NWP having best correlations with sea level? Do author have interpretations?
We speculate that since Japanese researchers made the COBESST-2 data, it is possible that there is improved data input or calibration for the region. As a result, this dataset appears to have more accurate results for NWP area. However, we have no studies to support this speculation, and merely state that it is the best case.

Why does not the local SST do better job than others?
We thought that SLA-KP is influenced larger scale variability that is also expressed outside of the immediate region of the KP. The best reconstruction will likely result when considering the domain that most nearly captures the physical processes impacting sea level around the KP. It is found that this domain is larger than the immediate area around the KP but smaller than a global domain.

Also, the short names e.g ReSLA-NWP are not used in figures, which however use the long name. Authors need to be careful for the presentation throughout the whole paper.
We have checked the shortened variables and corrected them.

Again in Fig 14 & 15, are the linear trend and annual cycle both removed before calculating correlation?
Yes, we removed trend before calculating correlation coefficient.

Are the trends in Fig 14b statistically significant?
Most of the trends are statistically significant. But some of them are not. We have added the p-test result.

[Figure]

'these detailed fluctuations are closer to the actual sea level variability': what is the actually sea level variability?

In this case, 'the actual sea level variability' means AVISO-KP – basically the sea level coming from actual observations. We have clarified this in the manuscript.

Authors seem to insist that the SST-based reconstruction shows better results. What are the reasons for that?
The main reasons that this reconstruction shows better results are 1) the use of CSEOF, 2) the incorporation of SST, and 3) the domain consideration. As we mentioned in the paper, the other reconstructions were not specifically focused on the SLA-KP. In a global reconstruction, there is a high possibility of omitting some important modes in certain local scale reconstruction. Using the CSEOFs, we are also attempting to account for periodic behavior that is difficult to capture with EOFs, and while also trying to represent potential lagged relationships between each basis functions.

In the marginal seas of NWP, many studies have shown that the local ocean surface currents and wind tress determine the sea level, and the open ocean in far-field has less impacts. However, this paper finds the (far-field) NWP SST can 'statistically' better capture the sea level in marginal seas of NWP i.e. KP. What is the science behind it? Please keep in mind that the sea level variations between the two sides of western boundary currents (Kuroshio/Oyashio) are very differently forced e.g. by the thermalsteric height and open ocean currents via geostrophic balance and by local wind/surface currents.
This reconstruction was conducted simply by extending the PCTs of AVISO-KP CSEOF, using SST as a statistical proxy. As mentioned above, we cannot explain most of the CSEOF modes in a physical manner. Indeed, other reconstructions have the same problem. At beginning stage, we tried to understand the background mechanism and the relation between the factors which you mentioned above. This remains a challenge, particularly given the complicated dynamics that the reviewer correctly points out. We have attempted to convey that SLA-KP is related with many factors: ocean currents, thermal expansion, global sea level rise, wind and so on, and it is expected SST-KP cannot cover each these factors. By ensuring that this variability is in some way captured by the CSEOF modes we use (even if we cannot say in which mode it resides), we should still be able to represent it in our resulting reconstruction.

More essentially, this paper is focusing on 'reconstruction capability', but it spent a lot space in section 3.1 comparing TG and AVISO. Authors should work properly to make the presentation and structure of this paper concise and focused.
We have omitted unnecessary parts.

Conclusion: What is the linear trend map of reconstructed SLA-KP over satellite era? Are they comparable with Fig 2?

[Figure]

We calculated the linear trend map of ReSLA-KP over the satellite era (1993-2014). Even a time period is not exactly same but the result agrees with the AVISO-KP's trend map.

How are the SST variations looking like over this region/NWP? Does SST follow the sea level changes very well?

We are unsure what the reviewer is exactly looking for with regards to an explanation of the SST variations of SST-NWP. But, as I showed you above, the relationship between SST and SLA is strong in terms of the CSEOF description. The figure below attempts to explain this process.

[Figure]

Minor comments. For example:

P2, lines 6-7: do not understand. What does bias mean?
It means that most of the TG stations are located on the Northern Hemisphere. We have clarified this.

Also, references are needed to support this statement. P2, lines 29-31: references?
Instead of reference, we have added global linear trend map.

P2 lines: 31-32: do not understand. P2 line 33: this needs to reword
We have rewritten this part.

Original:
Properly planning for future sea level change requires an assessment of sea level on local or regional levels, as future sea level for one location could be quite different than future sea level in another location. Rather than using a global reconstruction, several studies have instead focused on regional reconstructions of sea level, targeting a specific area of focus.

Modified
Therefore, it is necessary to assess the sea level changes in local or regional level for planning for future sea level accurately, as local sea level changes can have significant differences with the global sea level rise. Several studies have focused on regional reconstructions targeting a particular area of interest.

P3 lines 14-15: what reconstructions?
Global reconstructions by Hamlington et al (2011) and Church and White (2011)
We have rewrite this part.

P3 lines 11-22: references?
We have put the reference.

P3 lines 11-27: the focus/motivations are loosen and not concise.
We have rewritten this part.

P4 lines 9-10: do not understand
We have erased this sentence.

Figure 1 is not readable Figure 1 seems to have 3 TGs on China coasts, while there is only one appearing in Figure 3. Any flags applied?
The two TG's time spans did not cover 1993-2014.

---

## Author Response (AR2)

**Response to the comments (Referee 1)**

(a) The sea level change may be associated with many factors such as ocean temperature (including SST), salinity, currents, and surface winds etc. Therefore, the multi-regression between SST and SL PCTs may not include all aspects of SL changes. I am wondering whether the reconstruction could further be improved if more physical variables are considered.

To apply multi-variables to current reconstruction scheme, there are several problems. First, when we applied multi-variable's PCTs as predictor, the over-fitting occurred because as the mode goes higher, the possibility of overfitting increases. Second, some data that has less relation with SLA ruined the right signals. Third, the reanalysis process can increase the uncertainties of reconstruction. Actually, we tried to reconstruct SLA using wind and SST data simultaneously, but the result is poorer than each individual reconstruction case. To solve these problems, we need to input lots of efforts and we thought that is beyond our study boundary. However, the reconstruction applying multi-variables is valuable topic for the future study.

(b) The SL reconstruction does not include TG observations, but have a clear improvement over a similar reconstruction that includes TG observations. I am wondering whether the SL reconstruction could further be improved if all available TG observations are included.

To include TG data, there are two problems. First, using TG data we cannot conduct CSEOF analysis because they have lots of discontinuous points and their spatial coverage are too poor. So, we cannot establish the proper regression relationship between TG and SST. Second, the TG data's quality is not good. The vertical land motions cannot be calibrated and a lot of Japanese TG could be suffered by earthquakes or volcanic activities. To use TG data, the additional researches are necessary to correct the vertical land motions.

(c) How to validate the SL reconstruction in the early period over 1900-30 when no TG observations are available. It might be a little risky to include the reconstruction in this period.

As we can see in Fig. 11, even we have TG data for vilification, prior to 1970, the agreement between MSLA from TG and ReSLA-KP is very poor. But we thought that it does not mean our reconstruction is not good because the TG data is not enough to verify. Even we cannot verify the reconstruction results, we think the result is still valuable.

(d) Writing and presentation may need improving. There are too many abbreviations such as SL, MSL, GMSL, SL-KP. For example, MSL and GMSL could be explained in figure captions. KP is unnecessary because the study focuses on KP region only.

We have modified the abbreviations. But we cannot omit every KP, because, after removing KP, we found the unnecessary global terms must be necessary.

(e) Figure captions should identify the data source and average region etc.
We have applied the commend.

**Detailed comments**

P1L11, revise: extend the spatial resolution ..into the past
We have revised.

P3L5, CSEOF is not defined
We have defined in Page 2.

P3L14, "KP" could be deleted throughout the manuscript since the study has been limited over the KP region anyway, which will greatly improve the readability. "KP" could be noted in the figure caption when necessary.

We have deleted 'KP' and add explanation about the default domain.

P3L21, revise: looking at the regional level will lead to

We have fixed.

P4L11, annual signal=> seasonal signal?

The terms, annual signal and seasonal signal, are having same meaning. But to prevent confusions, we change seasonal signal to annual signal.

P4L20, include data => included data?

We have fixed it.

P4L21, over => from?

We have fixed it.

P6L5-7, revise the sentence

We have fixed.

P6L11, delete "in this case", "really" P6L12, independent of => independent from?

We have erased this sentence.

P9L3, How does "summing" actually do, arithmetic or squareroot?

We mean Root Sum Square. And this part have deleted.

P10L10, this is an indication that SL is not merely dependent on SST.

We have deleted this part. And we gave up to explain the physical reasons for the extreme SLR values.

P11L22, delete "then"

We have deleted it.

P11L25, delete "cases of"

We have deleted them.

P11L28-29, delete "considering the available number of TG data"

We have deleted them.

P12L4, It is not clear how MSLA-KP is defined (assuming every ocean grid in reconstruction). How MSLA-KP can be compared with TG-KP (only in TG grids).

We have added more explanations as follow.
To check the reconstruction results, we calculated MSLA of TG-KP, ReSLA-H, and ReSLA-KP. Spatial mean was calculated for the two grid datasets. For TG-KPs, we calculated mean differences between each time steps and we integrated the differences. The integrated mean differences became the MSLA of TG-KP.

P12L12, revise "was edited to have the same time span data gaps"

We have erased 'data gaps'.

P12L14-15, revise the sentence: ReSLA-KP show a better agreement of AVISO-KP than ReSLAH.
We have revised.

P12L17-18, how many modes are used in Hamlington?
We have added detailed number.
Hamlington et al. (2011) used a limited number (< 90% of total variance) of CSEOF modes to avoid over-fitting issues, but in this study, nineteen CSEOF modes are used which explain 98% of total variance of SLA-KP.

P12L23, thousand => a thousand
We have corrected.

P13L17-18, authors should extend the conclusion of a better current SL reconstruction. there is no way from Figures 16-17 to tell the current study is better.
We have deleted this part.

It is not clear in Figure 13 either. It may be necessary to point to Figure 14a. A better way is to calculate the RMSE.
We have added more figure.

Fig. 1, digital quality should be improved.
I think it has a high resolution, 600 ppi.

Fig. 2, coastal line should be consistent with those in other figures.
We have changed the figure.

Fig. 3, I could caption the figure as "Mean SLA in KP (gray) and global (black) regions from AVISO" so that I can get rid of some abbreviations.
We have changed the caption and figure.

Fig. 4, add "AVISO" in caption
We have modified.

Fig. 5, add "AVISO" in caption
We have modified caption.

Fig. 6, revise: trends (shapes) and correlation (color), change the red color of triangle into black so that the color will not be confused with correlation.
We have modified the caption and figure.

Fig. 7, NRMSE, I don't know the advantage of using normalized RMSE instead of RMSE.
'NRMSE' and 'RMSE' very similar, but when NRMSE has 'zero' value this means the regression is same with some constant value cases and if the value are negative that the compared data is less agreed its mean value. So I NRMSE gives some intuitive interpretation.

Figs. 8-9, I am confused how the 3-month averaged mode is plotted. I assume there is only one CSEOF associated with one PCT for a particular mode.
Yes, you are right. But the evolution is small through the months, therefore we represent the results as seasonal mean values to save some space.

Fig. 10, I assume this is for KP region
We have modified the caption.

Fig. 11, which region, KP region?
We have modified the caption.

Fig. 12, Why does Hamlington have a constant Corr and NRMSE?
Because we have 6 cases but ReSLA-H is just one case.

Fig. 14, "yellow" is barely identifiable. Why the correlation is over 1993-2008 while trend is over 1970-2008?
I think the color problem is related to the resolution, I provide 600 ppi image and that figure has no problem to recognize yellows.
And the time period is 1970-2008 for the both cases.

Fig. 15, The figure look great but there is a question: Since the study uses the CSEOF derived from AVISO, therefore validation against AVISO is considered to be not independent. One may argue that if authors use Hamlington deriving CSEOF, the performance reconstruction may be close to Hamlington.
Yes, that's right. Nevertheless, ReSLA-H has very poor agreement. We just want to show the limit of global reconstruction, as  you can see the below figure. Over 1993-2015, the correlation coefficient is values are pretty high, this means if we applied Hamlington et al. (2011)'s method in local scale, the correlation coefficients must be higher than current Fig.

[Figure]

**Response to the comments (Referee 2)**

1. Title and abstract: sea level projections seem to be an important aspect in this paper.
Unfortunately, there are no results related to 'projections'.
We admitted the title was not proper and I omitted the 'projection' from the title.

2. Introduction: authors believe that the sea level reconstruction using SST provides better results than the
conventional methods (using TGs). However, SST was also sparsely observed in early years including
ICOADS. How well do the SST methods cope with this common concern? Clarification is needed.
Actually, if we can secure the reasonable number of TG data, the reconstruction using TG data is the best
case. Unfortunately, TG data around the KP is less than 10 prior to mid 1960s. Therefore, we used SST
data instead of TG data. And to supplement the sparse observations, we made several cases: different
datasets, different areas.

I still believe the wind stress and local surface currents are dynamically important for sea level variations,
like many studies have shown. There is no direct link between coastal sea level and SST in open ocean.
How possible to include other dynamical factors?
SST and sea level has strong relationship when we analyze both of data through CSEOF analysis
(Hamlington et. al., 2011, 2012a). Actually, many CSEOF modes shows great agreement, e.g., ENSO and
PDO (see, below figures. LHS and RHS are ENSO modes of AVISO and OISST, respectively; c.f. we
just introduce these to show the example of their relationship)

[Figure]

As explained in 2.2.1 and 2.2.2, we tried to find the lagged relationship between PCts (SST and sea level).
Actually, only a few modes can be interpreted. Only we can understand that each mode is mathematically
orthogonal. This means these modes can be the best prediction variables. Therefore, even we cannot
explain the exact physical background of each modes and their relationship, the reconstruction results are
still valuable. And this kind of situation is same with other reconstruction studies.
And I had a reconstruction case that used wind data, but the result is not good as much as SST. (see below
figure). Because we cannot introduce every result and some cases were skipped.

[Figure]

3. Section 2: this part reads loosen and tediously long, and many parts are unnecessarily mentioned with many times. I would suggest shortening this section with concise contents to avoid readers losing interests.

We have trimmed out lots of repeating parts in sections.

4. Section 3: This section again is not properly presented.

We have modified this section.

4.1 Essential questions: 3.1: I do not think the following key question is answered. 'To reason whether the extreme trends patterns was related to the local mass distribution caused by various sources such as vortex and river discharge or was an independent. . ..' The extreme trends on China coasts in Fig 2 are proposed as a result of increasing river discharge by the authors. However, there is no convincing evidence supporting this. (one would not expect that river discharge can cause sea level increase on north Chinese coasts, because it is drying over recent years in this region). Same for the ocean current impacts. Can authors provide evidences supporting this (P10 Lines 28-31)?

We agree with your comment. There are no studies that explain the relation between the sea level rise trend and ocean current (or river discharge). To explain the relationship, we need more research that is beyond this paper's boundary. So, we removed this part.

Also, I cannot see any point of separating the regions with local correlations </>0.5. Because the two regions are both located in Yellow Sea and Japan Sea, the regional averages are supposed to not contain local information, and they instead reflect the large-scale variations. This might be reason why the two series in Fig5 are always highly correlated.

This study's basis assumption is the SLA-KP can be represent as the difference with the GMSL. And we worried about the extreme trend zones because if the extreme zones had significant differences with the other zone than the separated reconstructions were necessary. So to pull out the extreme zones we calculated the averaged correlation coefficient.

For the correlation map e.g. Fig4 (and Fig 6), is the annual cycle removed? Removing the seasonal cycle is critical. Otherwise, they are always statistically correlated but it does not make any sense. Need to clarify.

Yes, we removed the annual cycle. We have clarified it on the figure captions.

How can the sea level records between TG and AVISO be correlated e.g. Fig 6 when also having linear trends? If linear trends exist, they are always correlated. Correlation is for assessing the similarity between detrended variability/anomalies but cannot be used for assessing the trends. The basic concept I think is wrong.

We removed linear trends during the calculation of correlation coefficient. In my opinion, for sea level data the linear trend is very small variance than the data fluctuation so the trend has very less effect on the correlation coefficient values (see below figure). I checked the correlation coefficients of Fig 6 after removing linear trend of each time series. But it only made less than 1% changes.

[Figure]

The trends and correlation coefficients in Fig. 6, actually, calculated separately, there were several figures before. To reduce the figures, we combined the information and put the information in one figure. See the below figures.

[Figure]

Figure 3. SLA linear trends over 1993-2013 using TGs_KP

Figure 4. SLA linear trends over 1993-2013 using AVISO_sla

Figure 6. Correlation Coefficients between TGs_KP and AVISO_sla points with seasonal signals

Please clarify. Fig 6 & 7: how far are the AVISO sites from TG stations?
I clarified the maximum distance (about 12 km) at the caption.
In Fig. 7, but, to calculate MSLA of AVISO, I just used entire area not the closest point.

Fig8 & 9: I cannot see there is a trend in the PC series of Fig9.
We have changed the Fig.8 and 9 to show bigger PCT. We think you can see the trend.

What are the trend value and its significance level?
To help readers' understanding, we have added one more figure. Fig 12 shows the linear trend values of each mode and their confidence intervals.

Does it agree with the values based on the local estimation i.e. Fig 3.
Fig 3 doesn't have annual signal. If you see Fig 11, the wiggled signals can achieve by the summation of each CSEOF modes. So, our answer is Yes it does, it agree with MSLA-KP in terms of low-frequency signal.

Because there is no annual cycle signal in Fig9, there is no need of presenting it with 4 seasons.
The spatial pattern of one CSEOF analysis is not a single map, so they need to represent through their nested period. Actually, most of the CSEOF mode do not have similar spatial patterns though the time evolution. The reason why we can determine the 2$^{nd}$ mode as the trend mode, is these spatial patterns are pretty similar though the nested period.

4. 2 Section 3.2: what are the reasons for COBESST2-NWP having best correlations with sea level? Do author have interpretations?
We though the COBESST-2 data was made by Japanese researchers, and this means there is high possibility that the calibration can be focused on their near boundaries. So, this datasets have more accurate results for NWP area. But there are no similar studies before we cannot support this interpretations with reference. We just provide the best cases.

Why does not the local SST do better job than others?
We thought that SLA-KP is influenced by ocean current too. But the small domain's SST data is enough to interpret this ocean current effect. And the global SST contains too much information.

Also, the short names e.g ReSLA-NWP are not used in figures, which however use the long name. Authors need to be careful for the presentation throughout the whole paper.
I have checked the shorten variables and corrected them.

Again in Fig 14 & 15, are the linear trend and annual cycle both removed before calculating correlation?
Yes, we removed trend before calculating correlation coefficient.

Are the trends in Fig 14b statistically significant?
Most of the trends are statistically significant. But some of them are not. We have added the p-test result.

[Figure]

'these detailed fluctuations are closer to the actual sea level variability': what is the actually sea level variability?
'the actual sea level variability' means AVISO-KP. We have changed this.

Authors seem to insist that the SST-based reconstruction shows better results. What are the reasons for that?
As we mentioned in the paper, the other reconstructions were not focused on the SLA-KP and they didn't used entire decomposed mode for the reconstruction process. This means that the global reconstruction has high possibility to omit some important modes in certain local scale reconstruction. And also, for we applied lagged regression, we can include the lagged relationship between each basis functions.

In the marginal seas of NWP, many studies have shown that the local ocean surface currents and wind tress determine the sea level, and the open ocean in far-field has less impacts. However, this paper finds the (far-field) NWP SST can 'statistically' better capture the sea level in marginal seas of NWP i.e. KP. What is the science behind it? Please keep in mind that the sea level variations between the two sides of western boundary currents (Kuroshio/Oyashio) are very differently forced e.g. by the thermalsteric height and open ocean currents via geostrophic balance and by local wind/surface currents.
This reconstruction was conducted by extending the PCTs of AVISO-KP CSEOF. And, we cannot explain most of the CSEOF modes. The other reconstructions have the same problem. At beginning stage, we tried to understand the background mechanism and the relation between the factors which you mentioned above. But we figured out that our trials were beyond our research boundary. We thought that SLA-KP related with many factors: ocean currents, thermal expansion, global sea level rise, wind and so on, and SST-KP is not big enough to cover these factors. Even though SST-KP contains every effect, but the problem is whether CSEOF can decompose these factors well. And global SST contained too much information which can lead a over-fitting issue.

More essentially, this paper is focusing on 'reconstruction capability', but it spent a lot space in section 3.1 comparing TG and AVISO. Authors should work properly to make the presentation and structure of this paper concise and focused.
We have omitted unnecessary parts.

Conclusion: What is the linear trend map of reconstructed SLA-KP over satellite era? Are they comparable with Fig 2?

[Figure]

We calculated the linear trend map of ReSLA-KP over the satellite era (1993-2014). Even a time period is not exactly same but the result agrees with the AVISO-KP's trend map.

How are the SST variations looking like over this region/NWP? Does SST follow the sea level changes very well?

I don't know how I can explain the SST variations of SST-NWP. But, as I showed you above, the relationship between SST and SLA is close. And we want to explain our reconstruction process again. We used SST's CSEOF PCTs as predictors for the multivariable regression. And using the past SST PCT and the regression relation we extended the AVISO-KP's PCTs to the past. Therefore, the evolutions between SLA and SST are not necessary to be similar.

[Figure]

**Minor comments**

P2, lines 6-7: do not understand. What does bias mean?
It means that most of the TG stations are located on the Northern Hemisphere.

Also, references are needed to support this statement. P2, lines 29-31: references?
Instead of reference, we have added global linear trend map.

P2 lines: 31-32: do not understand. P2 line 33: this needs to reword
We rewrite this part.

P3 lines 14-15: what reconstructions?
Global reconstructions by Hamlington et al (2011) and Church and White (2011)
We have rewrite this part.

P3 lines 11-22: references?
We have put the reference.

P3 lines 11-27: the focus/motivations are loosen and not concise.
We have rewritten this part.

P4 lines 9-10: do not understand
We have erased this sentence.

Figure 1 is not readable Figure 1 seems to have 3 TGs on China coasts, while there is only one appearing in Figure 3. Any flags applied?
The two TG's time spans did not cover 1993-2014.

**Response to the comments (Topic editor)**

**Topic Editor Decision: Reconsider after major revisions** (14 Nov 2017) by John M. Huthnance
Comments to the Author:
Dear Authors
Thank-you for your revised manuscript. In due course I shall be sending it back to both referees who asked to see it again after "major revision". However, there are a few things which I would ask you to consider before that.

You have responded to (both) referees' question about using SST rather than other variables (e.g. wind, runoff, tide-gauge records). However, I think that a few sentences on this question should be included in the final manuscript so that it is "self-contained" and eventual readers do not have to search the discussion to find answers.
We inserted follows.
One of the unique characteristics of the current study is that we only used SST as a proxy of former SLA; other studies, however, used TG data or combined data (TG and SST). There are multiple reasons why we chose not to use TG data for the current reconstruction. The first reason is due to both the poor data coverage and the poor data quality. There are relatively few tide gauges extending into the past in our study area, and even fewer that are of high quality (i.e., unaffected by vertical land motion, with few gaps, free of non-physical jumps). The second reason, and related to the first, is that due to a methodological characteristic of the CSEOF analysis, a dataset that is free of gaps (temporally continuous) is needed. To satisfy this requirement, we are led to other gridded reconstruction or reanalysis products. There are many types of data that could potentially be used in our scheme (e.g. wind, ocean current, precipitation, atmospheric pressure). We used only SST for the following reasons. 1) SST and SLA have a distinct relationship when we analyze both of data through CSEOF analysis (Hamlington et al., 2011; Hamlington et al., 2012a; Hamlington et al., 2016) and Hamlington et al. (2012b) showed that SST could be a good proxy of SLA in this part of the ocean. 2) Limiting our analysis to SST reduces the possibility of overfitting in the regression scheme we use to reconstruct. As a final benefit of using SST, we can check against the available tide gauge data to provide an independent comparison to our reconstruction.

I also wonder whether referee 1 will be happy with the large number of remaining abbreviations.
We reduced abbreviations.

Here are a few more editorial-type details.
Page 3 lines 16-17. This sentence about the number of early tide gauges is not clear; are you saying that the first was in 1930 and this was the only one until 1950?

| Original | Changed |
|---|---|
| Second, the temporal coverage of the TG around the KP (TG-KP) started around 1930 when the only TG had been available by 1950; | Second, the temporal coverage of the TG around the KP (TG-KP) started around 1930 and only one TG was available until 1950; |

Page 3 line 19. Better ". . is proposing for the KP a new scheme . ."
Sorry, we want to keep the original sentence because our scheme can be applied for the other regions.

Page 6 line after (4). ". . and the $\varepsilon$ is random error. . ."
We have corrected it.

Page 8 line 2 Not "boundary". Maybe simply ". . two years maximum lag. Using . ."
We have fixed it.

Page 8 line 27. Omit "of the linear trend"?
We have fixed it.

Page 8 line 29 "separate" (spelling)
We have fixed it.
Page 9 lines 10-11. "five TGs showed acceptable accuracies". This seems to assume that all error is attributable to TGs. But the TG provide real and important data in the right place. It is the AVISO data that is in the wrong place.
We agree with your comment and we modified our expressions little bit.
The comparison showed that only five TGs having less than 30\% of differences with the AVISO-KP's linear trend. Eleven TGs showed more than 30\% of underestimation and twenty-one TGs had more than 30\% of over estimation. While there was disagreement between TG locations and AVISO grids, over than thirty percent of differences were significant.

Page 9 line 30 What are the "Six" reconstructions – please say clearly what they are.
We explained the six reconstructions in Sec. 2.1.2 and Figure 14. We have fixed little bit.
We made six reconstructions (Sec. 2.1.2 and Fig. 14), and the mean SLAs of six reconstructions showed a reasonable agreement with the mean SLA of TG-KP over 1965-2014.

Page 9 line 32. ". . there were only a few . ."
We have fixed.

Page 10 lines 19-24 (Monte-Carlo) This is much the same as page 8 lines 5-9 except including sea-level trend. Probably these lines could be merged in one place.
We have fixed.

Figure 2 caption. "gauge" (spelling, twice)
We have fixed.

Figure 3, 4, 5, 7, 8, 15 captions. Replace "w/o" – "without"?
We have fixed.

Figure 9 caption. "Cumulative variance of CSEOF modes . ."
We have fixed.

[revised manuscript text omitted]

---

## Author Response (AR4)

**Response to comments (Referee 1)**

P1, L16, delete "and are happening now"
We have deleted it.

P1, L20, revise "Expensive decision", "are already being made". Can you revise it like this: Important decisions have to be made in both economic and societal activities.
We have revised it.

P2, L1, after this TG => TG, hereafter
We have fixed it.

P2, L 4, "necessarily only", delete one of the words. "weighted towards" => mostly in the NH.
We have been fixed it.

P3, L28, delete "to center data"
We have fixed it.

P6, eq (6), how is d determined?
The Eq (6) has no d-term. Eq. (3) has d-term and it is nested period. The nested period represents a periodic characteristic of Loading Vectors. The determination process is entirely up to the user. Usually, however, we can determine the nested period by considering the periodicity of an original signal. This requires some physical intuition about the dataset at hand. For the reconstruction, the choice of nested period has little impact on the resulting reconstruction as long as is not too short or is chosen at a length that is particularly ill-suited for the strongest signals in the data. In this study, the most robust repeating signal is an annual signal. So, we can determine the d-term on a yearly basis (e.g., 12-months, 24-months).

P7, L7, delete "system"
It has been deleted.

P7, L9, it is confusing in the value of what in "if the value".
We have changed it as follows.
(before)
The threshold cross-correlation value did not have a sensitive effect on the regression if the value can select more than ten predictors;
(after)
The threshold cross-correlation value did not have a significant effect on the regression as long as the value was chosen to allow for at least ten predictors;

P7, eq (7), is (t-rou) a variable of PCT or a factor of multiplication? If it is a variable (most likely, based on equation (5)), the left side might be expressed as PCT(t). If it is a multiplication, you may have to explain why.
We have fixed as follows.

$$PCT_{SLA}^{(m,n)} = \beta_0^m + \sum_{i=1}^{K} \beta_i^m PCT_{SST}^{(m,n-\rho_i^m)} + \epsilon^{(m,n)} \tag{7}$$

where $PCT_{SLA}^{(m,n)}$ is the $n$-th component of the $m$-th PCT of AVISO's CSEOF and $PCT_{SST}^{(i,n)}$ is the $n$-th component of the $i$-th PCT of SST's CSEOF; $\rho_i^m$ is a lag of maximum correlation between the $i$-th predictor and the $m$-th target; $\beta_0^m$ and $\beta_i^m$ are represent regresstion constants and regression coefficients for the $m$-th target.

**P7, L18, revise "Then using .." as Using …, we can then extend …"**
We have revised it.

**P8, L19, "that we assume" => to assume that**
We have changed it.

**P8, L27, deemed => thought**
We have changed it.

**P9, L5, Are "Eleven TGs" the location of eleven TGs, or eleven TG observations? It reads like TG observations, but the TG observations cannot be underestimated or overestimated.**
We have revised it as follow.

data, the $\rho$ values were estimated and the mean value of the $\rho$ was about 0.72 (Fig. 4). In Fig. 4, the 11 TG stations (square shapes) estimated the linear trend at least 30% lower than the AVISO, while the 21 TG stations (diamond shapes) overestimated the trend by than 30%. To figure out the effect of these disagreements, the mean SLA of AVISO was compared with the TG's

**P9, L14, Fig. 6a**
We have fixed it.

**P9, L17, Fig. 6b**
We have fixed it.

**Figure 6, what is the unit of the amplitude?**
The amplitude has no unit.

**Figure 7, why the modes 1-2 are different from Figure 6?**
Actual signals of each mode can be recovered by multiplying of Loading Vector and its corresponding PC time series. These two figures are showing different things.

**P10, L3, delete "Normalized Root Mean Square Error;"**
We have deleted.

**P10, L24, shows better agreement => shows a better agreement**
We have corrected it.

**P10, L26, for the reconstruction process (in Hamlington?)**

We have revised this part.

1) using a greater number of target modes for the reconstruction process than previous studies (Hamlington et al., 2011, 2012a, b), 2) considering lagged relations between PCTs. Hamlington et al. (2011, 2012a, b) used a limited number (< 90% of total variance) of CSEOF modes to avoid over-fitting issues, but in this study, we used nineteen CSEOF modes which explain 98% of total variance of SLA-KP by using selective predictors. Further considering lagged relation between targets and predictors,

**P11, second paragraph, the comparison between the trend of 1900–2014 and 1993–2015 is not relevant. The trends over the same time period have to be used. This has to be revised.**

We tried to show the alleviation of the local difference in Fig 2a. In Fig. 2a, the difference between Max and Min linear trend is about 7 mm/yr but in Fig. 12a, the difference between Max and Min linear trend is about 0.7 mm/yr. We have revised it as follow.

and the maximum and minimum linear trends are about 2.1 mm/yr and 1.4 mm/yr, respectively (Fig. 12). The difference, about 0.7 mm/yr, between two extreme values of the reconstructed SLA is much less than the AVISO over 1993-2015 (about 7 mm/yr), particularly in the Yellow Sea, (Fig. 2 and 12). This alleviation means that the extended reconstruction period can

**P11, L13, altimeter data => altimeter data over 1993–2015.**

We have revised it.

**Figures, a period "." is needed at the end of the figure caption.**

We have added the period.

**Figure 2 caption: two regions: one with high correlation coefficient (red-colored area in (a)) and the other with low correlation coefficient (blue-colored area in (a)).**

Actually, Figure 2 is not the correlation map, but we have mentioned about the color.

**Figure 2.** (a) Linear trend map of sea level anomalies around the Korean Peninsula from AVISO without annual signal from 1993 to 2015 (the red-colored area has a greater linear trend than 3.0 mm/yr and the blue-colored area has a less linear trend); (b) Spatial mean time series of sea level anomalies around the Korean Peninsula (red) and global (blue) from AVISO without annual signal.

**Figure 6, move the figure legends to the middle of the figure?**

We have changed as follow.

[Figure]

Figure 10, optional: change yellow into red?
We have changed as follow.

[Figure]

Figure 11, move the figure legend to the middle of the figure?
We have changed it.

[revised manuscript text omitted]